# Understanding Domain Generalization: A Noise Robustness Perspective

**Rui Qiao and Bryan Kian Hsiang Low**
Department of Computer Science, National University of Singapore
`rui.qiao@u.nus.edu,lowkh@comp.nus.edu.sg`

## Abstract

Despite the rapid development of machine learning algorithms for domain generalization (DG), there is no clear empirical evidence that the existing DG algorithms outperform the classic empirical risk minimization (ERM) across standard benchmarks. To better understand this phenomenon, we investigate whether there are benefits of DG algorithms over ERM through the lens of label noise. Specifically, our finite-sample analysis reveals that label noise exacerbates the effect of spurious correlations for ERM, undermining generalization. Conversely, we illustrate that DG algorithms exhibit implicit label-noise robustness during finite-sample training even when spurious correlation is present. Such desirable property helps mitigate spurious correlations and improve generalization in synthetic experiments. However, additional comprehensive experiments on real-world benchmark datasets indicate that label-noise robustness does not necessarily translate to better performance compared to ERM. We conjecture that the failure mode of ERM arising from spurious correlations may be less pronounced in practice. Our code is available at `https://github.com/qiaoruiyt/NoiseRobustDG`

## 1 Introduction

A common assumption in machine learning is that the training and the test data are independent and identically distributed (i.i.d.) samples. In practice, the unseen test data are often sampled from distributions that are different from the one for training. For example, a camel may appear on the grassland during the test time instead of the desert which is their usual habitat (Rosenfeld et al., 2020). However, models such as overparameterized deep neural networks are prone to rely on spurious correlations that are only applicable to the train distribution and fail to generalize under the distribution shifts of test data (Hashimoto et al., 2018; Sagawa et al., 2019; Arjovsky et al., 2019). It is crucial to ensure the generalizability of ML models, especially in safety-critical applications such as autonomous driving and medical diagnosis (Ahmad et al., 2018; Yurtsever et al., 2020).

Numerous studies have attempted to improve the generalization of deep learning models regarding distribution shifts from various angles (Zhang et al., 2017; 2022; Nam et al., 2020; Koh et al., 2021; Liu et al., 2021a; Yao et al., 2022; Gao et al., 2023). An important line of work addresses the issue based on having the training data partitioned according to their respective domains/environments. For example, cameras are placed around different environments (near water or land) to collect sample photos of birds. Such additional environment information sparks the rapid development of *domain generalization* (DG) algorithms (Arjovsky et al., 2019; Sagawa et al., 2019; Krueger et al., 2021). The common goal of these approaches is to discover the *invariant* representations that are optimal in loss functions for all environments to improve generalization by discouraging the learning of spurious correlations that only work for certain subsets of data. However, Gulrajani & Lopez-Paz (2020) and Ye et al. (2022) have shown that no DG algorithms clearly outperform the standard empirical risk minimization (ERM) (Vapnik, 1999). Subsequently, Kirichenko et al. (2022) and Rosenfeld et al. (2022) have argued that ERM may have learned good enough representations. These studies call for understanding whether, when, and why DG algorithms may indeed outperform ERM.

In this work, we investigate under what conditions DG algorithms are superior to ERM. We demonstrate that the success of some DG algorithms can be ascribed to their *robustness to label noise* under subpopulation shifts. Theoretically, we prove that when using overparameterized models trained with

finite samples under label noise, ERM is more prone to converging to suboptimal solutions that mainly exploit spurious correlations. The analysis is supported by experiments on synthetic datasets. In contrast, the implicit noise robustness of some DG algorithms provides an extra layer of performance guarantee. We trace the origin of the label-noise robustness by analyzing the optimization process of a few exemplary DG algorithms, including IRM (Arjovsky et al., 2019), V-REx (Krueger et al., 2021), and GroupDRO (Sagawa et al., 2020), while algorithms including ERM and Mixup (Zhang et al., 2017) without explicit modification to the objective function, do not offer such a benefit. Empirically, our findings suggest that the noise robustness of DG algorithms can be beneficial in certain synthetic circumstances, where label noise is non-negligible and spurious correlation is severe. However, in general cases with noisy real-world data and pretrained models, there is still no clear evidence that DG algorithms yield better performance at the moment.

*Why should we care about label noise?* It can sometimes be argued that label noise is unrealistic in practice, especially when the amount of injected label noise is all but small. However, learning a fully accurate decision boundary from the data may not always be possible (e.g., some critical information is missing from the input). Such kinds of inaccuracy can be alternatively regarded as manifestations of label noise. Furthermore, a common setup for analyzing the failure mode of ERM assumes that the classifier utilizing only the invariant features is only partially predictive (having <100% accuracy) of the labels (Arjovsky et al., 2019; Sagawa et al., 2020; Shi et al., 2021). These setups are subsumed by our setting where there initially exists a fully predictive invariant classifier for the noise-free data distribution, and some degree of label noise is added afterward. More broadly, we analyze the failure mode of ERM without assuming the spurious features are generated with less variance (hence less noisy) than the invariant features. By looking closer at domain generalization through the lens of noise robustness, we obtain unique understanding of when and why ERM and DG algorithms work.

As our main contribution, we propose an inclusive theoretical and empirical framework that analyzes the domain generalization performance of algorithms under the effect of label noise:

1. We theoretically demonstrate that when trained with ERM on finite samples, the tendency of learning spurious correlations rather than invariant features for overparameterized models is jointly determined by both the degrees of *spurious correlation* and *label noise*. (Section 4)

2. We show that several DG algorithms possess the noise-robust property and enable the model to learn invariance rather than spurious correlations despite using data with noisy labels. (Section 5)

3. We perform extensive experiments to compare DG algorithms across synthetic and real-life datasets injected with label noise but unfortunately find no clear evidence that noise robustness necessarily leads to better performance in general. To address this, we discuss the difficulties of satisfying the theoretical conditions and other potential mitigators in practice. (Section 6, 7)

## 2 RELATED WORK

**Understanding Domain Generalization.** It has been demonstrated that ERM can fail in various scenarios. When spurious correlation is present, ERM tends to rely on spurious features that do not generalize well to other domains because of: model overparameterization (Sagawa et al., 2020), geometric failure from the max-margin principle (Nagarajan et al., 2020; Chaudhuri et al., 2023), feature noise (Khani & Liang, 2020), gradient starvation (Pezeshki et al., 2021), etc. Those setups assume that the invariant features are either partially or fully predictive of the labels. By incorporating label noise, our analysis applies to a broader range of settings. We also explicitly quantify the effect of spurious correlation along with label noise and provide a more intuitive understanding.

**Algorithms for Domain Generalization.** Myriads of algorithms and strategies have been proposed from different angles of the training pipeline, including 1) new training objectives by considering the domain/group information (Arjovsky et al., 2019; Sagawa et al., 2019; Krueger et al., 2021; Liu et al., 2021b; Zhang et al., 2021b; Ahuja et al., 2021; Zhou et al., 2022); 2) optimization by aligning the gradients (Koyama & Yamaguchi, 2020; Parascandolo et al., 2020; Shi et al., 2021; Rame et al., 2022); 3) efficient data augmentation (Zhang et al., 2017; Verma et al., 2019; Li et al., 2021; Yao et al., 2022; Gao et al., 2023); 4) strategically curated training scheme (Izmailov et al., 2018; Nam et al., 2020; Cha et al., 2021; Liu et al., 2021a; Zhang et al., 2022). However, multiple works have benchmarked that ERM and its simple variants can already achieve competitive performance across datasets (Gulrajani & Lopez-Paz, 2020; Idrissi et al., 2022; Khani & Liang, 2021; Ye et al., 2022; Chen et al., 2022; Yang et al., 2023; Liang et al., 2023). The quality of features learned by ERM

has also been assessed to be probably good enough (Kirichenko et al., 2022; Rosenfeld et al., 2022; Izmailov et al., 2022), challenging the practicality of various DG algorithms. Our work complements these studies with extensive experiments and discussions about the gap between theory and practice.

**Algorithmic Fairness** aims to remove the predictive disparity from groups (Dwork et al., 2012; Hardt et al., 2016) and its objectives are related to DG (Creager et al., 2021). Wang et al. (2021) analyze the harm of group-dependent label noise to models trained with fairness algorithms, while we show that uniform label noise hurts ERM but can be resisted by some DG algorithms.

## 3  DOMAIN GENERALIZATION WITH LABEL NOISE

Let $(\mathcal{X}, \mathcal{Y})$ denote the space of input and label, respectively. Let $\mathcal{E}$ represent the set of environments. Assuming that the environment information is known, denote a noise-free dataset by $D = \{(x^{(1)}, y^{(1)}, e^{(1)}), \ldots, (x^{(N)}, y^{(N)}, e^{(N)})\}$, where each data point $(x^{(i)}, y^{(i)}, e^{(i)}) \in \mathcal{X} \times \mathcal{Y} \times \mathcal{E}$ is sampled i.i.d. from a data distribution $\mathcal{P}$. For clarity, we will slightly abuse the notation by omitting $e^{(i)}$ from the data point when we only need $(x^{(i)}, y^{(i)})$. Each environment $e \in \mathcal{E}$ has its subset of data $D_e = \cup_{i:e^{(i)}=e}\{(x^{(i)}, y^{(i)})\}$ with the corresponding distribution $\mathcal{P}_e$. Let $f \in \mathcal{F}$ be the classifier in the hypothesis space $\mathcal{F}$ and $\ell$ be the loss function. The risk for the environment $e$ is defined as:

$$\mathcal{R}_e(f) = \mathbb{E}_{(x,y)\sim\mathcal{P}_e}[\ell(f(x), y)], \qquad \widehat{\mathcal{R}}_e(f) = (1/|D_e|) \sum_{(x,y)\in D_e} \ell(f(x), y).$$

Given the training environments $\mathcal{E}_{tr}$, the environment-balanced ERM estimator is:

$$\widehat{\mathcal{R}}^{\text{ERM}}(f) = \sum_{e\in\mathcal{E}_{tr}} \widehat{\mathcal{R}}_e(f).$$

The classic environment-agnostic ERM can be viewed as just having one environment in the training set. On the other hand, the objective of DG is to minimize the out-of-distribution (OOD) risk:

$$\mathcal{R}^{\text{OOD}}(f) = \max_{e\in\mathcal{E}} \mathcal{R}_e(f). \tag{1}$$

There are two major types of distribution shifts encountered in DG: *domain shifts* and *subpopulation shifts*. For domain shifts, the test environments generally have unseen populations with different compositions and features compared to the training environments. For example, cameras placed at different locations capture images for different populations with different backgrounds. Usually, there is no association between environments except that it is common to assume the existence of some domain-invariant feature that generalizes equally well to all training and test domains.

For subpopulation shifts, the training and the test environments share the same composition in terms of subpopulations, but the proportion of the subpopulations may vary. The subpopulations are also referred to as *groups*, denoted as $\mathcal{G}$. Note that the notion of *environment* is different from *group*. In general, group information, often associated with specific spurious features/attributes, is more challenging to obtain than environment information in practice. Let $g \in \mathcal{G}$ be an element of the groups and let $\mathcal{Q}_g$ be its associated data distribution. For subpopulation shifts, we can rewrite the data distribution for each environment $e$ as $\mathcal{P}_e = \sum_g \pi_g^e \mathcal{Q}_g$, where $\pi_g^e \in [0, 1]$ is the mixing coefficient and satisfies $\sum_g \pi_g^e = 1$. Thus, the OOD risk in Equation 1 can be rewritten as:

$$\mathcal{R}^{\text{OOD}}_{\text{subpop}}(f) = \max_{g\in\mathcal{G}} \mathcal{R}_g(f).$$

Therefore, we may evaluate the robustness to subpopulation shifts using the worst-group (WG) error in the test set. In this work, our theoretical results focus on subpopulation shifts, but we provide extensive empirical results for both types of distribution shifts.

Let the label-independent noise level be $\eta$, which is then applied to the noise-free dataset $D$ and generate the noisy dataset $\widetilde{D} = \{(x^{(1)}, \tilde{y}^{(1)}, e^{(1)}), \ldots, (x^{(N)}, \tilde{y}^{(N)}, e^{(N)})\}$, where $\tilde{y}^{(i)}$ is the result of randomly perturbing $y^{(i)}$ to another class with probability $\eta$. Let $\widehat{\mathcal{P}}$ be the corresponding noisy data distribution. Define the training risk of the classifier $f$ w.r.t. the noisy data distribution as:

$$\mathcal{R}^{\eta}(f) = \mathbb{E}_{(x,\tilde{y})\sim\widehat{\mathcal{P}}}[\ell(f(x), \tilde{y})], \qquad \widehat{\mathcal{R}}^{\eta}(f) = (1/N) \sum_{(x,\tilde{y})\in\widetilde{D}} \ell(f(x), \tilde{y}).$$

## 4  UNDERSTANDING THE EFFECT OF LABEL NOISE

Our analysis primarily focuses on subpopulation shifts. One common issue for subpopulation shifts is the strong *spurious correlations* between certain input features and the labels in the training

set, meaning that such spurious features can accurately predict the labels for the *majority* of the training data. This causes the trained classifier to rely heavily on the spurious features that do not generalize, especially to the *minority* groups that have the inverse of such correlation. Consequently, it is challenging to ensure the OOD performance when the test set is dominated by minority groups.

To obtain more intuitive theoretical results, we opt for the setup of binary linear classification with overparameterization *without* environment information (Sagawa et al., 2020; Nagarajan et al., 2020). For a training set $(\mathbf{X}, \mathbf{y})$ sampled from $\mathcal{P}$, we have $\mathbf{X} \in \mathbb{R}^{n \times d}$, $d \gg n$ for overparameterization, and $\mathbf{y} \in \{0, 1\}^n$. Consider that there exists a fine-grained and *disjoint* partition of the input features for each instance $\mathbf{x} = [\mathbf{x}_{\text{inv}}, \mathbf{x}_{\text{spu}}, \mathbf{x}_{\text{nui}}]$, where $\mathbf{x}_{\text{inv}}$ are invariant features that generalize across all environments $\mathcal{E}$, $\mathbf{x}_{\text{spu}} \in \mathbb{R}^{d_{\text{spu}}}$ are spurious features that only have a high *spurious correlation* $\gamma$ with label $y$ in the training environments $\mathcal{E}_{tr}$, and $\mathbf{x}_{\text{nui}} \in \mathbb{R}^{d_{\text{nui}}}$ are nuisance features sampled from a high-dimensional isotropic Gaussian distribution $\mathbf{x}_{\text{nui}} \sim \mathcal{N}(\mathbf{0}_{d_{\text{nui}}}, \sigma_{\text{nui}}^2 \mathbf{I}_{d_{\text{nui}}})$ and irrelevant to the task, but nonetheless provide linear separability for both the noise-free and the noisy training sets $D$ and $\tilde{D}$ due to overparameterization of the model and underspecification of finite samples (D'Amour et al., 2022). We use $\gamma \in (0.5, 1)$ exclusively for the correlation between spurious features and the label. The cardinalities of the features $d = d_{\text{inv}} + d_{\text{spu}} + d_{\text{nui}}$ and $d_{\text{nui}} \gg d_{\text{inv}} + d_{\text{spu}}$. This also resembles practical situations such as image classification where for the high-dimensional inputs, there are only a few invariant and spurious features that work for the classification task, but deep neural networks have high capacity and hence the ability to memorize using the nuisance features. Let the overparameterized linear classifier $f(\mathbf{x}) = \mathbf{w}^\top \mathbf{x}$, where the weights $\mathbf{w} \in \mathbb{R}^d$ can also be decomposed into $[\mathbf{w}_{\text{inv}}, \mathbf{w}_{\text{spu}}, \mathbf{w}_{\text{nui}}]$ w.r.t. $[\mathbf{x}_{\text{inv}}, \mathbf{x}_{\text{spu}}, \mathbf{x}_{\text{nui}}]$. We drop the bias term $b$ for simplicity.

**Assumption 4.1** (Linear Separability for Invariant Features). The noise-free data distribution $\mathcal{P}$ is linearly separable using just the invariant features $\mathbf{x}_{\text{inv}}$.

This assumption ensures that the invariant features are sufficiently expressive and have an accuracy that is at least as good as the spurious features for any environments or groups in a noise-free setting. When combined with label noise level $\eta \in [0, 0.5]$, this also accommodates the scenario with imperfect invariant classifiers. Thus, the assumption is in fact more inclusive than it is restrictive.

## 4.1 FAILURE-MODE ANALYSIS FOR ERM DUE TO LABEL NOISE

We first provide an analysis of the failure mode for ERM given a finite and noisy dataset *without* any environment information. Let $\mathbf{w}^{(\text{inv})}$ be the optimal classifier that only uses the invariant and nuisance features (i.e., the weights for spurious features $\mathbf{w}_{\text{spu}}^{(\text{inv})} = \mathbf{0}$, but not necessarily for the invariant component $\mathbf{w}_{\text{inv}}^{(\text{inv})}$ and nuisance component $\mathbf{w}_{\text{nui}}^{(\text{inv})}$). Denote the $\ell_2$-norm for the invariant component as $\|\mathbf{w}_{\text{inv}}^{(\text{inv})}\|$. Similarly, let $\mathbf{w}^{(\text{spu})}$ and $\|\mathbf{w}_{\text{spu}}^{(\text{spu})}\|$ be the optimal classifier that only uses the spurious and nuisance features (i.e., $\mathbf{w}_{\text{inv}}^{(\text{spu})} = \mathbf{0}$) and its associated $\ell_2$-norm. We assume that $\|\mathbf{w}_{\text{inv}}^{(\text{inv})}\| \geq \|\mathbf{w}_{\text{spu}}^{(\text{spu})}\|$. Since norms are usually the complexity measure of the models, this assumption conforms to the practical cases that invariant correlations are usually more complex and harder to learn compared to spurious correlations (recall the cow-camel example in Section 1). When ERM is the objective function, both types of classifiers $\mathbf{w}^{(\text{inv})}$ and $\mathbf{w}^{(\text{spu})}$ could reach the global optimum on the training set by learning the weights corresponding to the invariant/spurious features and memorizing all other noisy samples (because both classifiers achieve near 0 loss and gradient).

It has been demonstrated that a classifier optimized over ERM with gradient descent (GD) has the implicit bias of converging to a solution with minimum $\ell_2$-norm while minimizing the training objective (Zhang et al., 2021a), i.e., the *min-norm bias*. More details about this on logistic regression can be found in Appendix C.2. To show that the model does not rely only on the invariant features, it suffices to find a classifier that achieves near 0 loss with less norm than $\mathbf{w}_{\text{inv}}$. We then analyze *which type of classifier has a smaller norm* and is thus preferred by GD under the noisy and overparameterized linear classification setting. Due to the simplicity bias of deep learning (Geirhos et al., 2020; Shah et al., 2020) and abundant empirical observations, we assume that during training, the classifier first reaches a small-norm solution that only uses spurious features $\mathbf{x}_{\text{spu}}$. Thereafter, to further minimize the noisy empirical risk $\widehat{\mathcal{R}}^\eta(f)$, the classifier either (1) memorizes the noisy data, or (2) gradually picks up the invariant features to correctly classify the remaining noise-free non-spurious data (Arpit et al., 2017; Nagarajan et al., 2020) and potentially become $\mathbf{w}^{(\text{inv})}$. If $\mathbf{w}^{(\text{inv})}$ has a larger norm, it is harder for the model to escape the spurious solution using option (2).

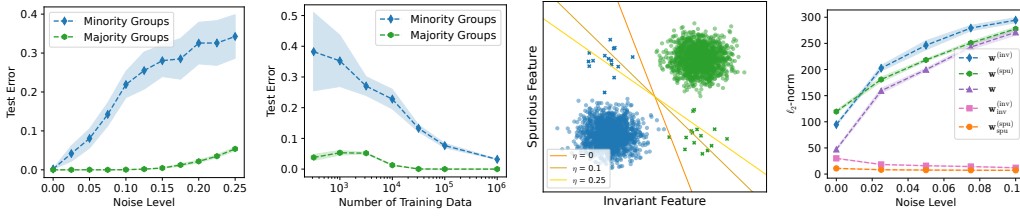

(a) Noise vs. test err    (b) Num of data vs. test err    (c) Decision boundaries    (d) Noise vs. weight norm

Figure 1: Simulation on synthetic data trained with overparameterized logistic regression. The dotted lines are the means and the shaded regions are for the standard error from 5 independent runs. Figure 1a shows that the minority-group (worst-group) error increases much more than the majority-group error when label noise is injected. Figure 1b indicates that gathering more data effectively reduces all test errors despite the presence of label noise. Figure 1c visualizes the learned decision boundaries for the same training set under different $\eta$, where markers "$\circ$" and "$\times$" are for the majority and the minority groups respectively. All data points are colored by the true labels. By adding more label noise, the classifier becomes more skewed towards using the spurious features. Figure 1d shows that as more noise is present, $\mathbf{w}^{(\mathrm{spu})}$ indeed tends to have a smaller norm than $\mathbf{w}^{(\mathrm{inv})}$, even though it is bigger when $\eta = 0$.

**Theorem 4.2.** *Under Assumption 4.1, if* $\|\mathbf{w}_{\mathrm{inv}}^{(\mathrm{inv})}\|^2 - \|\mathbf{w}_{\mathrm{spu}}^{(\mathrm{spu})}\|^2 \geq n(1-\gamma)(1-2\eta)C$, *then with high probability,* $\|\mathbf{w}^{(\mathrm{inv})}\| \geq \|\mathbf{w}^{(\mathrm{spu})}\|$ *, where* $C > 0$ *is a constant for memorization cost.*

For readability, we use $C$ to represent the average memorization cost. The proof is available in Appendix C.1. The theorem describes the condition required for the squared $\ell_2$-norm between the invariant component $\mathbf{w}_{\mathrm{inv}}^{(\mathrm{inv})}$ of $\mathbf{w}^{(\mathrm{inv})}$ and the spurious component $\mathbf{w}_{\mathrm{spu}}^{(\mathrm{spu})}$ of $\mathbf{w}^{(\mathrm{spu})}$, such that the $\ell_2$-norm of the desirable invariant classifier $\mathbf{w}^{(\mathrm{inv})}$ will be larger than the undesirable spurious classifier $\mathbf{w}^{(\mathrm{spu})}$. It implies that having a more severe spurious correlation $\gamma$ and higher noise level $\eta$ both make the condition in Theorem 4.2 easier to satisfy. By the min-norm bias of GD, $\mathbf{w}^{(\mathrm{spu})}$ having a lower norm than $\mathbf{w}^{(\mathrm{inv})}$ results in the overparameterized linear classifier to prefer exploiting the spurious correlation appearing in the training set and pay less attention to the invariant features, which hurts the generalization performance on minority groups in the test set. Besides, the theorem has intuitive interpretations of different factors. When the spurious correlation $\gamma \to 1$, the spurious features become invariant for the training environments, and the gap condition in Theorem 4.2 is satisfied ($\|\mathbf{w}_{\mathrm{inv}}^{(\mathrm{inv})}\|^2 - \|\mathbf{w}_{\mathrm{spu}}^{(\mathrm{spu})}\|^2 \geq 0$) so that this "newborn" invariant features with less norm would be favored. Furthermore, when the noise level $\eta \to 0.5$, there will be no useful features under this binary setting, and there is no difference between spurious features and invariant features anymore. As the gap condition is also satisfied, the spurious dummy classifier with the least norm is preferred. Lastly, when $n$ grows, the gap condition is enlarged and becomes harder to satisfy. This means that having more data could gradually resolve the harm from both spurious correlation and label noise.

To further illustrate the result, we study the effect of label noise on toy data trained with overparameterized logistic regression under the same setting. For each class in $\{0, 1\}$, the data that can be classified correctly by only $\mathbf{x}_{\mathrm{spu}}$ forms a majority group, and those that cannot belong to the minority group. We let there be a 1-dimensional latent variable that controls the generation for invariant and spurious features respectively. We set $d_{\mathrm{inv}} = d_{\mathrm{spu}} = 5$, $\gamma = 0.99$, $n = 1000$, $d_{\mathrm{nui}} = 3000$. The setup is modified from Sagawa et al. (2020) and more details are available in Appendix D.

As shown in Figure 1a, despite ERM generalizing to minority groups in the test set almost perfectly when it is trained with high spurious correlation on noise-free data, gradually adding noise degrades the minority (worst-case) performance significantly. Moreover, label noise increases the classifier's dependence on the spurious features, as the decision boundary becomes more tilted in Figure 1c. This complements the previous analyses that attribute the failure mode of ERM to the geometric skew (Nagarajan et al., 2020) of the max-margin classifier in the noise-free setup. Our study shows that adding label noise also intensifies the skew. Since the classification error for the minority group goes up gradually, one interpretation is that adding more noise gradually reduces the number of effective data points from the minority groups for invariance learning. Consequently, sampling more data

could help mitigate the effect of label noise and reduce test error on the minority groups, and we verify that by adding more data shown in Figure 1b. Thus, data collection and augmentation are still the keys to improving domain generalization. In addition, we train the classifiers $\mathbf{w}^{(\mathrm{inv})}$ and $\mathbf{w}^{(\mathrm{spu})}$ and plot the related $\ell_2$-norms w.r.t. increasing noise level in Figure 1d. As the noise level goes up, $\mathbf{w}^{(\mathrm{spu})}$ indeed tends to have a smaller norm than $\mathbf{w}^{(\mathrm{inv})}$. At the same time, the change in norm gap between $\mathbf{w}^{(\mathrm{spu})}_{\mathrm{spu}}$ and $\mathbf{w}^{(\mathrm{inv})}_{\mathrm{inv}}$ is relatively small. Moreover, the classifier $\mathbf{w}$ that uses all features also gets closer to the spurious classifier $\mathbf{w}^{(\mathrm{spu})}$. It also hints that $\ell_2$-regularization may not help with ERM for better robustness to subpopulation shifts when the label noise is present.

## 5 THE IMPLICIT NOISE ROBUSTNESS OF DG ALGORITHMS

We analyze a few exemplary baseline algorithms used for DG on their noise robustness. In particular, invariance learning (IL) algorithms such IRM and V-REx have better label-noise robustness by providing more efficient cross-domain regularization. We show the analysis for IRM and leave the rest in Appendix B. Let the classifier be $f = w \circ \Phi$, where $w$ is linear and $\Phi$ is the features/logits extracted from the input $x$. The practical surrogate for IRM objective on the training set is:

$$\widehat{\mathcal{R}}^{\mathrm{IRMv1}} = \sum_{e \in \mathcal{E}_{tr}} \widehat{\mathcal{R}}_e(w \circ \Phi_e) + \lambda \|\nabla_{w|w=1.0} \widehat{\mathcal{R}}_e(w \circ \Phi_e)\|^2 ,$$

where $\lambda$ is the regularization hyperparameter. In particular, IRM aims to learn representations that are local optima for all environments in addition to minimizing the sum of all risks. Suppose that these representations are invariant. Intuitively, since the noisy data need to be memorized independently, each of such costly memorization is non-invariant across domains and induces extra penalties on the regularization terms for domain invariance. For binary classification trained with the logistic loss (equivalent to the cross-entropy loss), the gradient for IRMv1 can be written as:

$$\nabla_\theta \widehat{\mathcal{R}}^{\mathrm{IRMv1}} = (1 + 2\lambda\phi[\sigma(\phi) + \phi\sigma(\phi)(1 - \sigma(\phi)) - y]) \nabla_\theta \widehat{\mathcal{R}}_e = \alpha(\phi)\nabla_\theta \widehat{\mathcal{R}}_e ,$$

where $\phi$ is the predictive logit for an instance $x$, $\sigma$ is the sigmoid function, and $\nabla_\theta \widehat{\mathcal{R}}_e$ is the ERM gradient w.r.t. the model parameters. As we have rewritten the IRM gradient as ERM gradient multiplied by a coefficient $\alpha(\phi)$, we can compare the two optimization trajectories directly. We plot the coefficient function $\alpha$ w.r.t. logit $\phi$ in Figure 2. Since $\alpha$ is symmetric w.r.t. $y = 0$ and $y = 1$, we only need to analyze for $y = 1$. During training, $\phi$ grows for $y = 1$ with ERM gradient. However, when $\phi$ is larger than a certain threshold (inversely proportional to the regularization strength $\lambda$), $\alpha(\phi)$ becomes negative, which reverses the gradient direction and does the opposite of minimizing the ERM loss by decreasing $\phi$. This behavior with strong regularization generally prevents the model from predicting training instances with high probabilities and memorizing the noisy samples with overconfidence. When the regularization for IRM is sufficiently strong ($\lambda > 100$ in practice), the loss for a single sample would oscillate back and forth instead of saturating.

This is part of the reason that we focus on analyzing the gradients of the algorithms trained on finite samples instead of performing convergence analysis. The gradient for V-REx has a similar property and we leave the analysis in Appendix B.2. On the other hand, the objectives for ERM, Mixup, and GroupDRO do not explicitly penalize memorization. As long as the model capacity allows, memorizing noisy data always leads to lower training loss and is encouraged by gradient descent. Thus, they are likely to have poorer noise robustness and worse generalization. However, since GroupDRO prioritizes optimizing the worst group, it generally slows down noise memorization in the short horizon. We will verify these claims in Section 6.

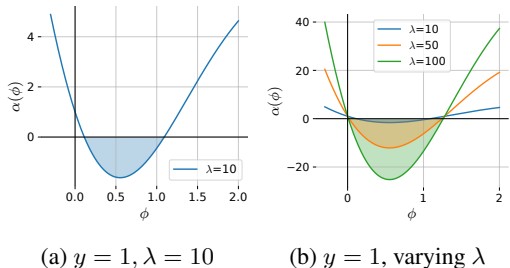

(a) $y = 1, \lambda = 10$     (b) $y = 1$, varying $\lambda$

Figure 2: IRMv1 gradient coefficient function $\alpha$ w.r.t. regularization strength $\lambda$. The shaded area represents $\alpha(\phi) < 0$. As $\lambda$ increases, the valley below 0 also deepens, providing stronger resistance.

## 6 IS NOISE ROBUSTNESS NECESSARY IN PRACTICE?

We empirically investigate how label noise affects OOD generalization across various simulated and real-life benchmarks. We adopt the Domainbed framework (Gulrajani & Lopez-Paz, 2020) with

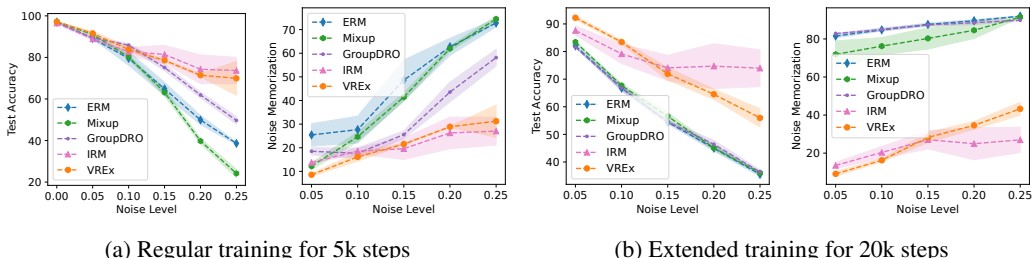

|  (a) Regular training for 5k steps | (b) Extended training for 20k steps |

Figure 3: Simulation on CMNIST dataset. As the noise level $\eta$ increases, both IRM and V-REx exhibit better generalization and noise robustness compared to approaches with ERM objectives.

standard configurations and add varying levels of label noise $\eta$ to the training environments of the datasets. We assume the availability of a small validation set from the test distribution for model selection. For a fair and comprehensive comparison, Domainbed implements 3 trials of 20 sets of hyperparameters, which may not be sufficient to reproduce the best results for all algorithms. We evaluate the robustness to label noise of the DG algorithms based on the average OOD accuracy on the test set of the best-performing models (selected by the validation set) across 3 trials.

## 6.1 DATASETS

*Subpopulation shifts (Synthetic).* **CMNIST** (ColoredMNIST) (Arjovsky et al., 2019) is a synthetic binary classification dataset based on MNIST (LeCun, 1998) with class label $\{0, 1\}$. There are four groups: green 1s $(g_1)$, red 1s $(g_2)$, green 0s $(g_3)$, and red 0s $(g_4)$. The color of digit has a high spurious correlation $\gamma$ with the label in the training set, where most of the instances are from $g_1$ and $g_4$, which can be viewed as the majority groups. There are two training environments $e_1, e_2$ with different degrees of spurious correlations $\gamma_1 = 0.9, \gamma_2 = 0.8$, meaning that using only color achieves 85% training accuracy. However, the minority groups $g_2, g_3$ dominates the test set with $\gamma_{\text{test}} = 0.1$.

*Subpopulation shifts (Real-world).* Waterbirds (Wah et al., 2011; Sagawa et al., 2019) is a binary bird-type classification dataset that has the label spuriously correlated with the image background. There are four groups: waterbirds on water $(\mathcal{G}_1)$, waterbirds on land $(\mathcal{G}_2)$, landbirds on water $(\mathcal{G}_3)$, and landbirds on land $(\mathcal{G}_4)$. The majority groups are $\mathcal{G}_1$ and $\mathcal{G}_4$, and the minority groups are $\mathcal{G}_2, \mathcal{G}_3$. Similarly, CelebA (Liu et al., 2015; Sagawa et al., 2019) is a binary hair color prediction dataset and blondness is much less correlated with the gender male than female, so blond male is the minority group and the rest are the majority. Based on these two datasets, we consider two types of simulated environments: (1) **Waterbirds** and **CelebA**: Directly treat the 4 groups as 4 environments, which is similar to the common setups, but we allow all algorithms including ERM to access the group information. However, this setup can be unfeasible when the group information is not available. (2) **Waterbirds+** and **CelebA+**: To simulate a more realistic setting, we create two derivative datasets based on Waterbirds and CelebA. There are two environments $e_1, e_2$. Each environment has half the data from each majority group. Then for each minority group, $1/3$ of the data is allocated to $e_1$ and the rest $2/3$ is allocated to $e_2$. This resembles the CMNIST setup so that $e_2$ has twice the amount of the minority data as $e_1$. This setting is less restrictive and the difference in the degree of spurious correlations represents a type of subpopulation shift. We also extend the result to the language modality using **CivilComments** (Borkan et al., 2019) in Appendix F.

*Domain shifts (Real-world).* Based on Domainbed (Gulrajani & Lopez-Paz, 2020), we use **PACS**, **VLCS**, **OfficeHome**, and **TerraIncognita**. Each dataset has multiple environments. Each environment has specific populations that have unique features that are different from the ones in other environments. This is distinct from subpopulation shifts where only the composition of the population changes across environments. We leave the full descriptions of all datasets in Appendix E.1.

## 6.2 EXPERIMENTS ON SYNTHETIC CMNIST DATA

For CMNIST, we train simple Convolutional Neural Network (CNN) models for 5k steps. Only models at the last epoch are compared for validation performance (i.e., no early stopping).

As shown in Figure 3a, when the noise level $\eta$ is close to 0, ERM performs competitively even without using the environment label information compared to other DG algorithms. Since on average $\gamma = 0.85$ in the training set, the degree of spurious correlation is not severe enough to make ERM fail in the noise-free case. However, when the noise level goes up, IL algorithms (IRM, V-REx) significantly outperform ERM-based algorithms (ERM, Mixup). The performance gap is enlarged when the noise level $\eta$ gets higher because the accuracy of ERM-based algorithms deteriorates much faster. On the other hand, GroupDRO is slightly behind IRM and V-REx but has a clear advantage over ERM. Moreover, we track the accuracy of the noisy data as a surrogate of noise memorization in Figure 3a, which shows that DG algorithms naturally are more robust to label noise memorization compared to vanilla ERM-based approaches. Furthermore, a strong negative correlation can be observed between noise memorization and the test error.

To further examine noise memorization, we "overfit" CMNIST dataset with 25% label noise by extended training for 20k steps in Figure 3b. IL algorithms such as IRM, V-REx with strong invariance regularization do not memorize the noisy data severely, despite the complete memorization still being a valid local minimum (all environments have 0 and equal loss). On the other hand, ERM, Mixup, and GroupDRO suffer from noise memorization in the long horizon. The phenomenon of GroupDRO having contrasting results when trained with different steps aligns with the previous conclusion that GroupDRO works well with strong regularization such as early stopping (Sagawa et al., 2019).

## 6.3 EXPERIMENTS ON REAL-WORLD DATA

We add 10% and 25% label noise to the benchmark datasets and compare the out-of-distribution performance for various algorithms. For all datasets, we use ResNet-50 (He et al., 2016) pretrained on ImageNet (Deng et al., 2009). For the real-world subpopulation-shift datasets, we report the worst-group (WG) accuracy of the model on the test set selected according to the validation WG performance. For the domain-shift datasets, we perform single-domain cross-test experiments by setting each domain as the test set and report the average accuracy of the model selected using 20% of the test data. Standard data augmentation is performed. More details are available in Appendix E.

| Dataset | Waterbirds+ | | Waterbirds | | CelebA+ | | CelebA | |
|---|---|---|---|---|---|---|---|---|
| Alg / $\eta$ | 0 | 0.1 | 0 | 0.1 | 0 | 0.1 | 0 | 0.1 |
| ERM | $81.5_{\pm0.4}$ | $64.6_{\pm0.5}$ | $86.7_{\pm1.0}$ | $61.4_{\pm1.3}$ | $71.7_{\pm2.9}$ | $65.4_{\pm1.9}$ | $88.1_{\pm1.2}$ | $\mathbf{63.1}_{\pm1.1}$ |
| Mixup | $79.8_{\pm0.9}$ | $61.8_{\pm2.0}$ | $\mathbf{88.0}_{\pm0.4}$ | $56.9_{\pm0.9}$ | $71.1_{\pm1.2}$ | $64.6_{\pm1.8}$ | $86.9_{\pm0.5}$ | $61.7_{\pm1.8}$ |
| GroupDRO | $\mathbf{82.6}_{\pm0.4}$ | $65.6_{\pm0.8}$ | $84.9_{\pm0.7}$ | $\mathbf{61.9}_{\pm0.7}$ | $71.9_{\pm2.0}$ | $61.7_{\pm1.7}$ | $87.6_{\pm0.4}$ | $63.0_{\pm1.4}$ |
| IRM | $78.5_{\pm0.8}$ | $63.3_{\pm0.5}$ | $87.0_{\pm1.4}$ | $58.8_{\pm0.8}$ | $\mathbf{75.4}_{\pm4.3}$ | $62.4_{\pm2.5}$ | $88.0_{\pm0.4}$ | $59.3_{\pm1.7}$ |
| V-REx | $78.3_{\pm0.3}$ | $\mathbf{66.6}_{\pm0.9}$ | $87.4_{\pm0.6}$ | $59.2_{\pm1.5}$ | $73.9_{\pm0.7}$ | $\mathbf{69.9}_{\pm2.0}$ | $\mathbf{89.3}_{\pm0.7}$ | $60.2_{\pm2.0}$ |

Table 1: Worst-group accuracy (%) for subpopulation shifts.

| Dataset | PACS | | VLCS | | OfficeHome | | TerraIncognita | |
|---|---|---|---|---|---|---|---|---|
| Alg / $\eta$ | 0.1 | 0.25 | 0.1 | 0.25 | 0.1 | 0.25 | 0.1 | 0.25 |
| ERM | $82.0_{\pm0.5}$ | $74.5_{\pm0.6}$ | $75.0_{\pm0.3}$ | $\mathbf{71.9}_{\pm0.6}$ | $62.2_{\pm0.1}$ | $54.9_{\pm0.3}$ | $52.1_{\pm0.5}$ | $48.5_{\pm0.3}$ |
| Mixup | $\mathbf{83.6}_{\pm0.1}$ | $\mathbf{75.2}_{\pm0.9}$ | $\mathbf{75.5}_{\pm0.2}$ | $\mathbf{71.9}_{\pm0.5}$ | $\mathbf{63.9}_{\pm0.1}$ | $\mathbf{57.5}_{\pm0.3}$ | $\mathbf{53.2}_{\pm1.2}$ | $\mathbf{51.3}_{\pm1.0}$ |
| GroupDRO | $82.4_{\pm0.3}$ | $74.7_{\pm0.4}$ | $75.1_{\pm0.1}$ | $71.2_{\pm0.2}$ | $61.3_{\pm0.4}$ | $54.1_{\pm0.3}$ | $51.8_{\pm0.7}$ | $50.6_{\pm0.6}$ |
| IRM | $80.4_{\pm1.2}$ | $71.0_{\pm1.9}$ | $74.6_{\pm0.3}$ | $70.3_{\pm0.3}$ | $61.2_{\pm1.2}$ | $53.9_{\pm1.8}$ | $48.3_{\pm1.4}$ | $44.4_{\pm2.1}$ |
| VREx | $81.4_{\pm0.2}$ | $73.5_{\pm0.7}$ | $75.0_{\pm0.1}$ | $71.8_{\pm0.6}$ | $60.6_{\pm0.5}$ | $53.0_{\pm0.9}$ | $48.3_{\pm0.3}$ | $48.9_{\pm0.4}$ |

Table 2: Cross-test accuracy (%) for domain shifts.

For real-world subpopulation-shift datasets in Table 1, the ranking of different algorithms varies across different datasets and noise levels. Despite being prone to spurious correlation and label noise, ERM has decent performance given the environments. The superiority of IL algorithms on CMNIST does not translate into real-world datasets. For real-world domain-shift datasets, adding label noise degrades the performance of all tested algorithms on a similar scale. Surprisingly, Mixup outperforms all other algorithms consistently, followed closely by ERM. This indicates the effectiveness of data augmentation for OOD generalization under label noise. Despite being noise-robust, IRM and

VREx rarely catch up with ERM. Thus, there is no clear evidence that the noise robustness property necessarily leads to better performance in real-world subpopulation-shift and domain-shift datasets. In practice, the choice of DG algorithms depends heavily on the dataset and hyperparameter search. An effective model selection strategy could be more important than an algorithm.

For Waterbirds and CelebA, our GroupDRO results are much lower than the reproducible baseline (Sagawa et al., 2019), showing that the current hyperparameter search space can be far from complete. However, by utilizing the group labels, our results for ERM, IRM, and V-REx are already significantly higher than the reported scores from prior works (Yao et al., 2022), setting *new baseline records*. Given that ERM and DG algorithms perform similarly under the current setting, it may be possible that ERM can be improved further by better model selection and broader hyperparameter search.

## 7 DISCUSSION

**Why does ERM objective perform competitively on real-world subpopulation-shift datasets?**
(1) The real-world training is usually coupled with model pretraining and data augmentation. With higher-quality representations and more augmented data, the learning requires much fewer training steps, which prevents noise memorization. We observe that even with strong spurious correlation $\gamma > 0.95$, noise memorization is less severe for all algorithms including ERM on Waterbirds(+) dataset in Appendix H. (2) The failure condition based on spurious correlation and label noise may not be satisfied in practice. Without knowing the group labels, the Waterbirds+ and CelebA+ datasets both exhibit strong spurious correlations between some input features and the labels. However, it is unclear how much more difficult it is to learn the invariant features compared to the spurious ones, which relates to the norm difference between the spurious and invariant classifiers. For example, in CelebA, the label hair blondness is spuriously correlated with the feature gender, but the gender feature could be more difficult to learn than the hair blondness, unlike CMNIST where the spurious feature color is more straightforward for the model to discover. Thus, the issue of spurious correlation on real-world datasets may be less severe. (3) The invariance learning condition may not be satisfied. IL algorithms require training environments to have sufficient distributional differences for the subpopulations to distinguish invariant features. This condition might be true for CMNIST ($\gamma_1 - \gamma_2 = 0.1$), but not necessarily for Waterbirds+ and CelebA+ with a less-than-2% difference in $\gamma$.

**Does spurious correlation affect domain shifts?** In domain shifts, it is common to assume there are domain-specific features that have spurious correlations with the labels. However, these spurious correlations picked up from the training environments are unlikely to be present in new domains. If this correlation is not so strong, we expect some level of meaningful representations can be learned by ERM and yield reasonable performance for test distributions. Moreover, when there are multiple training domains available that come with non-trivial spurious correlations, all the domain-specific spurious correlations combined may result in even more complexity and become a worse burden for the classifier to learn compared to the invariant features. We hypothesize that adding data from more domains is likely to encourage invariant feature learning even for just ERM.

## 8 CONCLUSION AND FUTURE WORK

Label noise constitutes another failure mode for ERM by exacerbating the effect of spurious correlations. Invariant learning algorithms exhibit a clear advantage of noise robustness over ERM-based algorithms in certain synthetic circumstances. Unfortunately, such desirable property does not translate to better generalization in real-world situations. ERM objective with effective data augmentation still constructs a simple yet competitive baseline. We believe that more theoretical and empirical analyses are needed to understand when and why ERM and DG algorithms work and fail.

In practice, it is uncertain whether the invariant and spurious features can be perfectly extracted. Subsequently, the invariant features may violate the linear separability. As such, our theoretical analysis may not be directly applicable in the real world. A more general understanding can be obtained by removing these assumptions. Furthermore, though the spurious correlations may be non-trivial to learn for some datasets, certain types of adversarial attacks may create similar conditions as in corrupted CMNIST, where invariance learning might provide stronger resilience. The implicit noise robustness of IL algorithms could also be beneficial to learning with noisy labels.

## 9 ACKNOWLEDGEMENT

We would like to thank the anonymous reviewers and AC for the constructive and helpful feedback. We thank Zheyuan Hu for the helpful discussions. This research/project is supported by the National Research Foundation, Singapore under its AI Singapore Programme (AISG Award No: AISG2-PhD/2021-08-017[T]).

## 10 REPRODUCIBILITY STATEMENT

For our theoretical analysis, we have stated the assumptions in Section 4 and given the proofs in Appendix C. For our empirical results, we have documented the complete experimental setup in Appendix E. We have also submitted our code with the necessary instructions to reproduce our results.

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

## A  DG ALGORITHMS

In this work, we will analyze some representative algorithms including Invariance Learning (IL) (Arjovsky et al., 2019; Krueger et al., 2021) and Distributionally Robust Optimization (DRO) (Sagawa et al., 2019). Let $f = w \circ \Phi$, where $w$ is linear and $\Phi$ is the features extracted from the input $x$. Let $\mathcal{E}_{tr}$ be the training environments. For IL, two popular risk objectives are:

$$\widehat{\mathcal{R}}^{\text{IRM}} = \sum_{e \in \mathcal{E}_{tr}} \widehat{\mathcal{R}}_e(w \circ \Phi) \qquad \text{s.t. } w \in \arg\min_{\bar{w}} \widehat{\mathcal{R}}^e(\bar{w} \circ \Phi), \ \forall e \in \mathcal{E}_{tr}$$

$$\widehat{\mathcal{R}}^{\text{V-REx}} = \sum_{e \in \mathcal{E}_{tr}} \widehat{\mathcal{R}}_e(f) + \lambda \mathbb{V}_{e \in \mathcal{E}_{tr}}[\widehat{\mathcal{R}}_e(f)]$$

where $\mathbb{V}[\widehat{\mathcal{R}}_e(f)]$ is the variance of risk across environments and $\lambda$ is the hyperparameter. For DRO, one of the risk objectives is the worst-case loss over all training environments:

$$\widehat{\mathcal{R}}^{\text{Group-DRO}} = \max_{e \in \mathcal{E}_{tr}} \widehat{\mathcal{R}}_e(f). \tag{2}$$

## B  NOISE ROBUSTNESS ANALYSIS

### B.1  GRADIENT ANALYSIS FOR IRM ALGORITHM

We show this by analyzing the gradient of IRM for finite samples instead of doing convergence analysis. Recall that the IRM objective function is:

$$\mathcal{R}^{\text{IRM}} = \sum_{e \in \mathcal{E}_{tr}} \mathcal{R}_e(w \circ \Phi) \qquad s.t. \ w \in \arg\min_{\bar{w}} R^e(\bar{w} \circ \Phi), \ \forall e \in \mathcal{E}_{tr}$$

The goal of IRM is to obtain classifiers that are simultaneously optimal across all training environments, but this induces a bilevel optimization problem that is hard to optimize. As zero gradient is necessary for reaching the local minimum, IRM has the practical surrogate that minimizes the norm of the gradient to encourage reaching the local minimum for all training environments. Since our goal is to understand how the DG algorithms work in practice, we only analyze the practical surrogate of IRM, which is the de facto implementation across standard benchmarks. The practical surrogate called IRMv1 has the form:

$$\mathcal{R}^{\text{IRM}} = \sum_{e \in \mathcal{E}_{tr}} \mathcal{R}_e(w \circ \Phi_e) + \lambda \|\nabla_{w|w=1.0} \mathcal{R}_e(w \Phi_e)\|^2 \tag{3}$$

For each environment $e$:

$$\mathcal{R}_e(w \circ \Phi_e) = \frac{1}{n} \sum_{i}^{n} \mathcal{R}_i(w \circ \Phi_i) \tag{4}$$

The gradient w.r.t. $w = 1.0$ for an environment $e$ is a scalar that can be rewritten as:

$$\nabla_{w|w=1.0} \mathcal{R}_e(w\Phi) = \left(\frac{\partial \mathcal{R}_e}{\partial \Phi}\right)\Phi$$

$$= \frac{1}{n} \sum_{i}^{n} \frac{\partial \mathcal{R}_i}{\partial \Phi_i} \Phi_i$$

For simplicity, we focus on IRMv1 and consider the binary classification case with logistic regression. The input $x \in \mathbb{R}^d$ and the label $y \in \{0, 1\}$. We analyze with logistic regression due to its simple form and most of the analysis can be done with scalars. Moreover, it is widely known that logistic regression is equivalent to training with cross-entropy loss, so there is no loss of generality. Logistic regression has the form:

$$p(y = 1|x) = \sigma(\theta^\top x),$$

where $\theta$ are the weights of the linear classifier, and $\sigma$ is the sigmoid function:

$$\sigma(x) = \frac{1}{1 + e^{-x}}.$$

Let $\phi = \theta^\top x \in \mathbb{R}$, which corresponds to $\Phi$ in our previous formulations. The IRM-ready logistic regression is simply:

$$p(y = 1|x) = \sigma(w\phi),$$

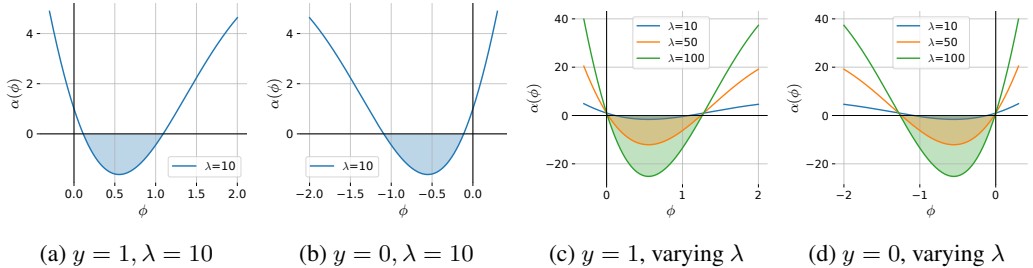

(a) $y = 1, \lambda = 10$     (b) $y = 0, \lambda = 10$     (c) $y = 1$, varying $\lambda$     (d) $y = 0$, varying $\lambda$

Figure 4: IRM gradient coefficient w.r.t. regularization strength $\lambda$

where $w = 1$ and we will drop it in the following analysis unless it is needed in the term. The IRM-ready logistic loss for an environment:

$$L(\theta) = -y \log \sigma(w\phi) - (1 - y) \log(1 - \sigma(w\phi)).$$

The partial gradient w.r.t. linear weights $\theta$:

$$\frac{\partial L(\theta)}{\partial \theta} = (\sigma(\phi) - y)x.$$

The partial gradient for $w$ used by IRMv1:

$$\nabla_{w|w=1} L(\theta) = (\sigma(\phi) - y)\phi.$$

Let $L_e$ be the logistic loss for environment $e$. The IRMv1 objective is:

$$L^{\text{IRM}} = \sum_e L_e(\theta) + \lambda \|\nabla_{w|w=1} L_e(\theta)\|^2.$$

For simplicity, we just analyze the behavior of IRMv1 for a single environment, so we abuse the notation and the subscript $e$ can be dropped temporarily. The gradient for IRMv1 in a *single* environment for one sample $(x, y)$ is:

$$
\begin{aligned}
\frac{\partial L_e^{\text{IRM}}}{\partial \theta} &= \frac{\partial L_e(\theta)}{\partial \theta} + \frac{\partial \lambda \|\nabla_{w|w=1} L_e(\theta)\|^2}{\partial \theta} \\
&= (\sigma(\phi) - y)x + 2\lambda(\sigma(\phi) - y)\phi[\sigma(\phi) + \phi\sigma(\phi)(1 - \sigma(\phi)) - y]x \\
&= (\sigma(\phi) - y)x(1 + 2\lambda\phi[\sigma(\phi) + \phi\sigma(\phi)(1 - \sigma(\phi)) - y]) \\
&= (1 + 2\lambda\phi[\sigma(\phi) + \phi\sigma(\phi)(1 - \sigma(\phi)) - y]) \frac{\partial L_e(\theta)}{\partial \theta} \\
&= \alpha(\phi) \frac{\partial L_e(\theta)}{\partial \theta}
\end{aligned}
$$

We have re-expressed the gradient of IRM in terms of the gradient of the original logistic function multiplied by a new coefficient function $\alpha$. This new coefficient $\alpha(\phi)$ for the gradient has an interesting property that restricts the predictive probability to non-extreme values when the regularization strength $\lambda$ is sufficiently large. We plot the coefficient function $\alpha$ w.r.t. $\phi$ in Figure 4. The function is symmetric with respect to label $y$. Without loss of generality, we only look at the case for $y = 1$. Recall that for logistic regression, we predict positive if $\phi > 0$. To minimize the logistic loss for an instance $(x_i, y_i)$ where $y_i = 0$, $\phi$ would keep growing because it increases $p(y_i|x_i)$. The coefficient $\alpha$ needs to stay positive for gradient descent. However, as $\phi$ increases, $\alpha$ becomes negative before $\phi$ is large enough, which blocks the logistic loss to continue decreasing and severe memorization. Moreover, a larger $\lambda$ leads to a stronger resistance during the optimization process as shown in Figure 4. The same rationale applies to the $y = 0$ case due to the symmetry. In practice, $\lambda$ (usually $> 100$) tends to be very large in order to work well on datasets like CMNIST. Thus, for IRM, data points may be oscillating between gradient descent and gradient ascent (w.r.t. the ERM case), and we do observe that IRM loss typically does not converge on the best-performing models. We hypothesize that only classification patterns that work for most of the data could cross the hill and get preserved.

### B.2 GRADIENT ANALYSIS FOR V-REX ALGORITHM

Recall that the V-REx loss function is:

$$\mathcal{R}^{\text{V-REx}} = \frac{1}{|\mathcal{E}_{tr}|} \sum_{e \in \mathcal{E}_{tr}} \mathcal{R}_e(f) + \lambda \mathbb{V}_{e \in \mathcal{E}_{tr}}[\mathcal{R}_e(f)]$$

$$= \frac{\lambda}{m} \sum_{e=1}^{m} (\mathcal{R}_e(f) - \bar{\mathcal{R}})^2 + \frac{1}{m} \sum_{e=1}^{m} \mathcal{R}_e(f) \, ,$$

where $\bar{\mathcal{R}} = (1/m) \sum_{e \in \mathcal{E}_{tr}} \mathcal{R}_e$ is the average loss of all training environments. For a specific environment $i \in \mathcal{E}_{tr}$, we can rewrite the loss as:

$$\mathcal{R}^{\text{V-REx}} = \frac{\lambda}{m} \sum_{e=1}^{m} (\mathcal{R}_e(f) - \bar{\mathcal{R}})^2 + \frac{1}{m} \sum_{e=1}^{m} \mathcal{R}_e(f)$$

$$= \frac{\lambda}{m} (\mathcal{R}_i(f) - \bar{\mathcal{R}})^2 + \frac{\lambda}{m} \sum_{e=1, e \neq i}^{m} (\mathcal{R}_e(f) - \bar{\mathcal{R}})^2 + \frac{1}{m} \mathcal{R}_i(f) + \frac{1}{m} \sum_{e=1, e \neq i}^{m} \mathcal{R}_e(f) \, .$$

The gradient incurred from $\mathcal{R}_i$ can be derived as follows:

$$\frac{\partial \mathcal{R}_i^{\text{V-REx}}}{\partial \theta} = \frac{\partial \mathcal{R}^{\text{V-REx}}}{\partial \mathcal{R}_i} \frac{\partial \mathcal{R}_i}{\partial \theta}$$

$$= [\frac{2\lambda}{m} (\mathcal{R}_i(f) - \bar{\mathcal{R}})(1 - \frac{1}{m}) + \frac{2\lambda}{m} \sum_{e=1, e \neq i}^{m} (\mathcal{R}_e(f) - \bar{\mathcal{R}})(-\frac{1}{m}) + \frac{1}{m}] \frac{\partial \mathcal{R}_i}{\partial \theta}$$

$$= [\frac{2\lambda}{m} (\mathcal{R}_i(f) - \bar{\mathcal{R}}) + \frac{2\lambda}{m} \sum_{e=1}^{m} (\mathcal{R}_e(f) - \bar{\mathcal{R}})(-\frac{1}{m}) + \frac{1}{m}] \frac{\partial \mathcal{R}_i}{\partial \theta}$$

$$= [\frac{2\lambda}{m} (\mathcal{R}_i(f) - \bar{\mathcal{R}}) + \frac{1}{m}] \frac{\partial \mathcal{R}_i}{\partial \theta}$$

$$= \frac{1}{m} [2\lambda(\mathcal{R}_i(f) - \bar{\mathcal{R}}) + 1] \frac{\partial \mathcal{R}_i}{\partial \theta} \, .$$

The overall gradient for the network parameters $\theta$ can be expressed as:

$$\frac{\partial \mathcal{R}^{\text{V-REx}}}{\partial \theta} = \sum_{e \in \mathcal{E}_{tr}} \frac{\partial \mathcal{R}_e^{\text{V-REx}}}{\partial \theta}$$

$$= \sum_{e \in \mathcal{E}_{tr}} \frac{1}{m} [2\lambda(\mathcal{R}_e(f) - \bar{\mathcal{R}}) + 1] \frac{\partial \mathcal{R}_e}{\partial \theta} \, .$$

In practice, the strength of regularization for invariance learning tends to be very large ($\lambda > 100$) for the better-performing models. For environments with below-average risk where $\mathcal{R}_e < \bar{\mathcal{R}}$, The coefficient will become negative, and gradient *ascent* instead of descent is performed for this batch of data points. We can view such gradient ascent as an "unlearning" process. We hypothesize that since memorization is a gradual process and only decreases the risk for its own environment, making it below average, its effect can be purged later on by gradient ascent just as other spurious features. As a result, only the invariant features that are useful to all environments are preserved.

## C PROOFS

### C.1 PROOF OF THEOREM 4.2

We first look at the memorization cost for overfitting the noisy data points.

**Lemma C.1.** *For an overparameterized linear classification problem with label noise, with high probability, the squared norm of the classifier $\|\mathbf{w}\|^2$ will be increased by $\beta^2 d_{\text{nui}} \sigma_{\text{nui}}^2$ on average for memorizing a single sample (mislabeled or non-spuriously correlated) in the training set, where $\beta \geq 0$ is a constant.*

*Proof.* First, by the definition of $\ell_2$-norm:

$$\|\mathbf{w}\|^2 = \|\mathbf{w}_{\text{inv}}\|^2 + \|\mathbf{w}_{\text{spu}}\|^2 + \|\mathbf{w}_{\text{nui}}\|^2 . \tag{5}$$

Since $\mathbf{w}_{\text{inv}}$ and $\mathbf{w}_{\text{spu}}$ can both result in relatively low classification error, memorization will mostly incur a larger norm on the term $\|\mathbf{w}_{\text{nui}}\|^2$.

Recall that $\mathbf{x}_{\text{nui}} \sim \mathcal{N}(0, \sigma_{\text{nui}}^2)$ and $d_{\text{nui}} \gg n$. Vectors sampled from high-dimensional Gaussian spheres are orthogonal to each other with high probability: $\forall i \neq j$, $\mathbf{x}_{\text{nui}}^{(i)} \perp \mathbf{x}_{\text{nui}}^{(j)}$. To memorize a set of data points $S$ using only the nuisance features, it is then sufficient to construct $\mathbf{w}_{\text{nui}}^{(S)} = \sum_{i \in S} \beta y^{(i)} \mathbf{x}_{\text{nui}}^{(i)}$, where $\beta \to 0$ and more importantly:

$$\|\mathbf{w}_{\text{nui}}^{(S)}\|_2^2 \approx \beta^2 \sum_{i \in S} \|\mathbf{x}_{\text{nui}}^{(i)}\|_2^2 . \tag{6}$$

By Bernstein's inequality for the norm of subgaussian random variables:

$$P\left(\left|\|\mathbf{x}_{\text{nui}}^{(i)}\|^2 - \mathbb{E}[\|\mathbf{x}_{\text{nui}}^{(i)}\|^2]\right| \geq t\right) \leq 2\exp\left(-\frac{cd_{\text{nui}}\min(t^2,t)}{K^4}\right)$$

$$P\left(\left|\|\mathbf{x}_{\text{nui}}^{(i)}\|^2 - d_{\text{nui}}\sigma_{\text{nui}}^2\right| \geq t\right) \leq 2\exp\left(-\frac{cd_{\text{nui}}\min(t^2,t)}{K^4}\right) ,$$

where $K = \max_j \|\mathbf{x}_{\text{nui}}^{(i)}\|_{\psi_2}$ and $c$ is an absolute constant. Thus, with high probability, $\|\mathbf{w}_{\text{nui}}^{(S)}\|^2$ concentrates on $\beta^2 |S| d_{\text{nui}} \sigma_{\text{nui}}^2$, and the average cost of memorizing a data point is $\beta^2 d_{\text{nui}} \sigma_{\text{nui}}^2$. In particular, the probabilistic bound becomes tighter as $d_{\text{nui}}$ increases, which corresponds to the model becoming more overparameterized. $\square$

We first compare the purely spurious classifier $\mathbf{w}^{(\text{spu})}$ and the purely invariant classifier $\mathbf{w}^{(\text{inv})}$ under the effect of noise memorization. Let $[\mathbf{a}; \mathbf{b}]$ be the concatenation of vector $\mathbf{a}$ and $\mathbf{b}$ vertically. Let $L$ be the logistic loss. The classifiers can be defined mathematically as:

$$\mathbf{w}^{(\text{spu})} = \arg\min_{[\mathbf{w}_{\text{spu}}; \mathbf{w}_{\text{nui}}]} \frac{1}{N} \sum_i L(y^{(i)}, [\mathbf{w}_{\text{spu}}; \mathbf{w}_{\text{nui}}]^\top [\mathbf{x}_{\text{spu}}^{(i)}; \mathbf{x}_{\text{nui}}^{(i)}]) .$$

$$\mathbf{w}^{(\text{inv})} = \arg\min_{[\mathbf{w}_{\text{inv}}; \mathbf{w}_{\text{nui}}]} \frac{1}{N} \sum_i L(y^{(i)}, [\mathbf{w}_{\text{inv}}; \mathbf{w}_{\text{nui}}]^\top [\mathbf{x}_{\text{inv}}^{(i)}; \mathbf{x}_{\text{nui}}^{(i)}]) .$$

**Theorem C.2.** *If* $\|\mathbf{w}_{\text{inv}}^{(\text{inv})}\|^2 - \|\mathbf{w}_{\text{spu}}^{(\text{spu})}\|^2 \geq n(1-\gamma)(1-2\eta)\beta^2 d_{\text{nui}}\sigma_{\text{nui}}^2$, *then* $\|\mathbf{w}^{(\text{inv})}\| \geq \|\mathbf{w}^{(\text{spu})}\|$.

*Proof.* With label noise level $\eta$, the number of data points that need to be memorized using the optimal set of invariant features is:

$$\tilde{n}_{\text{inv}} = n\eta .$$

The number of data points that need to be memorized using the $\gamma$-correlated spurious feature is:

$$\tilde{n}_{\text{spu}} = n(1 - \gamma + (2\gamma - 1)\eta) .$$

Then approximately,

$$\|\mathbf{w}^{(\text{inv})}\|^2 = \|\mathbf{w}_{\text{inv}}^{(\text{inv})}\|^2 + \|\mathbf{w}_{\text{nui}}^{(\text{inv})}\|^2$$
$$\approx \|\mathbf{w}_{\text{inv}}^{(\text{inv})}\|^2 + n\eta\beta^2 d_{\text{nui}}\sigma_{\text{nui}}^2$$
$$\|\mathbf{w}^{(\text{spu})}\|^2 = \|\mathbf{w}_{\text{spu}}^{(\text{spu})}\|^2 + \|\mathbf{w}_{\text{nui}}^{(\text{spu})}\|^2$$
$$\approx \|\mathbf{w}_{\text{spu}}^{(\text{spu})}\|^2 + n(1 - \gamma + (2\gamma - 1)\eta)\beta^2 d_{\text{nui}}\sigma_{\text{nui}}^2 .$$

Then rewrite the gap as:

$$\|\mathbf{w}^{(\text{inv})}\|^2 - \|\mathbf{w}^{(\text{spu})}\|^2 = \|\mathbf{w}_{\text{inv}}^{(\text{inv})}\|^2 + n\eta\beta^2 d_{\text{nui}}\sigma_{\text{nui}}^2 - (\|\mathbf{w}_{\text{spu}}^{(\text{spu})}\|^2 + n(1 - \gamma + (2\gamma - 1)\eta)\beta^2 d_{\text{nui}}\sigma_{\text{nui}}^2)$$
$$= \|\mathbf{w}_{\text{inv}}^{(\text{inv})}\|^2 - \|\mathbf{w}_{\text{spu}}^{(\text{spu})}\|^2 + n\eta\beta^2 d_{\text{nui}}\sigma_{\text{nui}}^2 - n(1 - \gamma + (2\gamma - 1)\eta)\beta^2 d_{\text{nui}}\sigma_{\text{nui}}^2$$
$$= \|\mathbf{w}_{\text{inv}}^{(\text{inv})}\|^2 - \|\mathbf{w}_{\text{spu}}^{(\text{spu})}\|^2 - n(1 - \gamma + (2\gamma - 2)\eta)\beta^2 d_{\text{nui}}\sigma_{\text{nui}}^2$$
$$= \|\mathbf{w}_{\text{inv}}^{(\text{inv})}\|^2 - \|\mathbf{w}_{\text{spu}}^{(\text{spu})}\|^2 - n(\gamma - 1)(2\eta - 1)\beta^2 d_{\text{nui}}\sigma_{\text{nui}}^2$$
$$= \|\mathbf{w}_{\text{inv}}^{(\text{inv})}\|^2 - \|\mathbf{w}_{\text{spu}}^{(\text{spu})}\|^2 - n(1 - \gamma)(1 - 2\eta)\beta^2 d_{\text{nui}}\sigma_{\text{nui}}^2 .$$

Suppose that $\|\mathbf{w}^{(\mathrm{inv})}\| \geq \|\mathbf{w}^{(\mathrm{spu})}\|$, then:

$$\|\mathbf{w}_{\mathrm{inv}}^{(\mathrm{inv})}\|^2 - \|\mathbf{w}_{\mathrm{spu}}^{(\mathrm{spu})}\|^2 - n(1-\gamma)(1-2\eta)\beta^2 d_{\mathrm{nui}}\sigma_{\mathrm{nui}}^2 \geq 0$$

$$\|\mathbf{w}_{\mathrm{inv}}^{(\mathrm{inv})}\|^2 - \|\mathbf{w}_{\mathrm{spu}}^{(\mathrm{spu})}\|^2 \geq n(1-\gamma)(1-2\eta)\beta^2 d_{\mathrm{nui}}\sigma_{\mathrm{nui}}^2 \ .$$

By the above equation, it is easy to see that larger $\gamma$ or $\eta$ both lead to a smaller difference required between $\|\mathbf{w}_{\mathrm{inv}}^{(\mathrm{inv})}\|^2$ and $\|\mathbf{w}_{\mathrm{spu}}^{(\mathrm{spu})}\|^2$ to have $\|\mathbf{w}^{(\mathrm{inv})}\|^2 \geq \|\mathbf{w}^{(\mathrm{spu})}\|^2$. As our classifier is implicitly regularized to prefer a decision boundary with a smaller norm, the converged solution becomes more likely to rely on spurious correlation, resulting in significantly worse generalization performance. □

*Remark* C.3. It is possible that the classifier that only uses the spurious features may achieve a smaller norm by replacing the memorization of certain data points with invariant feature learning. This is likely to be true in practice. However, our objective here is to identify at least one suboptimal candidate classifier that is preferred by ERM to the invariant classifier. Moreover, the classifier with both spurious and partially invariant features will have an even smaller norm than $\|\mathbf{w}^{(\mathrm{spu})}\|^2$, resulting in the model preferring this model more than the purely spurious classifier and thus also not converging to the purely invariant classifier.

## C.2    THE INDUCTIVE BIAS TOWARDS MINIMUM-NORM SOLUTION

The weight norm is usually associated with the complexity of the model. Techniques such as $\ell_2$-regularization are often used to improve the generalization of linear models by introducing the bias towards smaller norm solutions. Moreover, Zhang et al. (2021a) hypothesized the implicit regularization effect of stochastic gradient descent (SGD) during model training. They show the result for linear regression:

**Lemma C.4.** *For linear regression, the model parameters optimized using gradient descent w.r.t. mean squared error converges to the minimum $\ell_2$-norm solution:*

$$\mathbf{w}^* = \arg\min_{\mathbf{w}}\|\mathbf{w}\|^2 \quad \text{s.t. } y = X\mathbf{w} \ .$$

For classification, we follow a similar rationale proposed by (Sagawa et al., 2020). Recall that as we have assumed overparameterization, the training data is linearly separable. Consider the hard-margin SVM:

$$\mathbf{w}^* = \arg\min_{\mathbf{w}}\|\mathbf{w}\|^2 \quad \text{s.t. } y^{(i)}(\mathbf{w}^\top\mathbf{x}^{(i)}) \geq 1 \ \forall i \ .$$

In this case, the max-margin solution corresponds to the minimum-norm solution. Thus, it suffices to show if the model converges to the max-margin solution. Soudry et al. (2018) and Ji & Telgarsky (2018) demonstrated that logistic regression trained with gradient descent converges in direction towards the max-margin solution:

**Theorem C.5** (Soudry et al. (2018)). *For any linearly separable data, gradient descent for logistic regression will behave as:*

$$\mathbf{w}(t) = \mathbf{w}^* \log t + \boldsymbol{\rho}(t) \ , \tag{7}$$

*where $t$ is the step. The residual grows at most as $\|\boldsymbol{\rho}(t)\| = O(\log\log t)$, and so it converges towards the direction of the max-margin solution $\mathbf{w}^*$:*

$$\lim_{t\to\infty}\frac{\mathbf{w}(t)}{\|\mathbf{w}(t)\|} = \frac{\mathbf{w}^*}{\|\mathbf{w}^*\|} \ .$$

However, the global minimum of unregularized logistic regression is not well-defined. To bridge this gap, we can look at $\ell_2$-regularized logistic regression with regularization hyperparameter $\lambda$:

$$\mathbf{w}_\lambda^* = \arg\min_{\mathbf{w}}\frac{1}{N}\sum_i L(y^{(i)}, \mathbf{w}^\top\mathbf{x}^{(i)}) + \frac{\lambda}{2}\|\mathbf{w}\|^2 \ , \tag{8}$$

where $L$ is the logistic loss. As $\lambda \to 0^+$, we can recover the unregularized logistic regression solution. Moreover, it also reflects the max-margin/min-norm solution, which has been proved by Rosset et al. (2003) in their Theorem 2.1:

$$\lim_{\lambda\to 0^+}\frac{\mathbf{w}_\lambda^*}{\|\mathbf{w}_\lambda^*\|} = \frac{\mathbf{w}^*}{\|\mathbf{w}^*\|}.$$

Thus, we can mimic unregularized logistic regression with $\lambda$ that is extremely small, so that the norm is *finite*. The linear model also converges in a finite amount of time when it is trained in practice.

These results demonstrate the minimum-norm inductive bias of (stochastic) gradient descent. This allows us to derive conclusions based on the comparison of norms between different types of classifiers such as the only using invariant features or spurious features.

## C.3 STANDARD ERM STILL WORKS IN EXPECTATION

**Proposition C.6.** *For a binary classification problem using softmax classifiers trained with cross-entropy loss function on infinite samples, if the label-independent noise level $\eta < 1/2$, there exists a $f^*$ that is Bayes optimal and also a global optimum that satisfies first-order stationarity.*

*Proof.* Recall that $(x, \tilde{y})$ denotes a data point that is subject to label noise with ratio $\eta$. Since the original label for $x$ is $y$, then we can use $1 - y$ to represent the flipped label since its binary classification with labels $0, 1$. For notational convenience, we can let $q$ be the one-hot vector for $y$. It has the same dimension 2 as $f(x)$. Let $h$ be the logits for input $x$ in the final layer before passing through softmax, meaning that $f(x) = \text{softmax}(h)$. The gradient of the model parameters for correctly labeled data points ($\tilde{y} = y$):

$$
\begin{aligned}
\nabla \ell^+ &= \nabla \ell(f(x), y) \\
&= -\nabla \sum_i q_i \log f(x)_i & \text{(cross-entropy loss)} \\
&= -\nabla \log f(x)_y & (q_y = 1) \\
&= -\frac{1}{f(x)_y} \nabla f(x)_y \\
&= -\frac{1}{f(x)_y} \nabla \text{softmax}(h)_y \\
&= -\frac{1}{f(x)_y} [f(x)_y (q - f(x))]^\top \nabla h \\
&= (f(x) - q)^\top \nabla h
\end{aligned}
$$

The gradient for mislabeled data points ($\tilde{y} \neq y$):

$$
\begin{aligned}
\nabla \ell^- &= \nabla \ell(f(x), 1 - y) \\
&= -\frac{1}{f(x)_{1-y}} \nabla f(x)_{1-y} \\
&= (f(x) - (1 - q))^\top \nabla h
\end{aligned}
$$

We can consider the two logits $h_0$, $h_1$ separately. Without loss of generality, consider $y = 1$:

$$
\begin{aligned}
\nabla \ell^+ &= (f(x) - q)^\top \nabla h \\
&= (f(x)_1 - 1) \nabla h_1 + (f(x)_0 - 0) \nabla h_0 \\
\nabla \ell^- &= (f(x) - (1 - q))^\top \nabla h \\
&= (f(x)_1 - 0) \nabla h_1 + (f(x)_0 - 1) \nabla h_0
\end{aligned}
$$

With noise level $\eta$, in expectation:

$$
\begin{aligned}
\mathbb{E}[\nabla \ell] &= (1 - \eta) \nabla \ell^+ + \eta \nabla \ell^- \\
&= (1 - \eta)[(f(x)_1 - 1) \nabla h_1 + (f(x)_0 - 0) \nabla h_0] + \eta[(f(x)_1 - 0) \nabla h_1 + (f(x)_0 - 1) \nabla h_0] \\
&= (f(x)_1 - 1 + \eta) \nabla h_1 + (f(x)_0 - \eta) \nabla h_0 \\
&= (f(x)_1 - 1 + \eta) \nabla h_1 + (1 - f(x)_1 - \eta) \nabla h_0 & (f(x)_0 + f(x)_1 = 1)
\end{aligned}
$$

Clearly, $f(x)_1 = 1 - \eta$ is the unique solution to setting the expected gradient as 0, if we do not constrain the value of $\nabla h_1, \nabla h_0$, reaching the first-order stationary point with convergence. Moreover, this is also a minimizer and the Bayes optimal classifier $f^*$ as it achieves the Bayes optimal error rate $\eta$. $\qquad\square$

Fundamentally, the label noise level $\eta$ is essentially the same as Bayes error rate in expectation. This shows that logistic regression classifiers trained with ERM are naturally noise-robust *in expectation*.

**Proposition C.7.** *Given the noise level $\eta < 1/2$, the Bayes optimal classifier remains at least as accurate as the classifier using only $\gamma$-correlated features, i.e., $R_{0\text{-}1}^\eta(f^*) \le R_{0\text{-}1}^\eta(f^\gamma)$.*

*Proof.* The Bayes optimal classifier using all features for the noisy data has 0-1 loss:

$$R_{0\text{-}1}^\eta(f^*) = R_{0\text{-}1}(f^*) + \eta$$

The optimal classifier using only a set of $\gamma$-correlated features has the following 0-1 loss:

$$R_{0\text{-}1}^\eta(f^\gamma) = (1-\gamma)(1-\eta) + \gamma * \eta = 1 - \gamma + (2\gamma - 1)\eta$$

We use proof by contradiction. Suppose that $R_{0\text{-}1}^\eta(f^*) > R_{0\text{-}1}^\eta(f^\gamma)$. Recall that $\gamma \in [0,1]$, then

$$R_{0\text{-}1}^\eta(f^*) > R_{0\text{-}1}^\eta(f^\gamma)$$
$$R_{0\text{-}1}(f^*) + \eta > 1 - \gamma + (2\gamma - 1)\eta$$
$$(2 - 2\gamma)\eta > 1 - \gamma - R_{0\text{-}1}(f^*)$$
$$\eta > \frac{1}{2} - \frac{R_{0\text{-}1}(f^*)}{2 - 2\gamma} \qquad\qquad (1 - \gamma > 0)$$
$$\eta > \frac{1}{2} \qquad\qquad (R_{0\text{-}1}(f^*) = 0)$$

As $\eta \in [0, 0.5)$, the last inequality cannot be satisfied so the reverse $R_{0\text{-}1}^\eta(f^*) \le R_{0\text{-}1}^\eta(f^\gamma)$ must be true. $\square$

According to Proposition C.7, at least in expectation, the global minimum is still obtained by the Bayes optimal classifier even under the presence of label noise. However, in practice with only a finite number of samples, ERM (w/ or w/o regularization) could fail to learn a generalizable classifier when the noise level is high because of the joint effort between the low-dimensional spuriously correlated features and noise in the overparameterized setting. With limited samples, the training error refuses to converge easily to the Bayes optimal error since it can be further reduced by the memorization of noisy samples. In general, increasing the number of data would gradually resolve the issues of both label noise and spurious correlation (Figure 1b).

## D  SYNTHETIC EXPERIMENTS

Our synthetic experiments are based on the framework developed in Sagawa et al. (2020). Without loss of generality, we let $y \in \{-1, +1\}$ instead of $\{0, 1\}$ for the clarify of describing the data-generating process. Let there be a spurious attribute $a \in \{-1, +1\}$ with spurious correlation $\gamma \in [0.5, 1]$, such that in the training set, we have $p(a = y) = \gamma$. Subsequently, four groups are created: $(g_1)$: $y = -1, a = -1$; $(g_2)$: $y = -1, a = 1$; $(g_3)$: $y = 1, a = -1$; $(g_4)$: $y = 1, a = 1$. Let there be $n$ training data points. The number of data in each group $|g_1| = |g_4| = \gamma n/2$, $|g_2| = |g_3| = (1-\gamma)n/2$. The data is sampled as follows:

$$\mathbf{x}_\text{inv} \sim \mathcal{N}(y\mathbf{1}, \sigma_\text{inv}^2 \mathbf{I}_{d_\text{inv}}), \qquad \mathbf{x}_\text{spu} \sim \mathcal{N}(a\mathbf{1}, \sigma_\text{spu}^2 \mathbf{I}_{d_\text{spu}}), \qquad \mathbf{x}_\text{nui} \sim \mathcal{N}(\mathbf{0}, \sigma_\text{nui}^2 \mathbf{I}_{d_\text{nui}}/d_\text{nui}),$$

In this case, we set $d_\text{inv} = d_\text{spu}$ and $\sigma_\text{inv}^2 = \sigma_\text{spu}^2$ so that the two features are expected to have the same complexity and noise level. For the main plots (Figure 1a,1b) w.r.t. noise level $\eta$, we set $\gamma = 0.99$, $n = 1000$, $d_\text{nui} = 3000$, $d_\text{inv} = d_\text{spu} = 5$, $\sigma_\text{inv}^2 = \sigma_\text{spu}^2 = 0.25$. We fit the model with logistic regression with regularization $\lambda = 10^{-4}$. For each configuration (by noise level $\eta$ or number of data $n$), we repeat the experiments 10 times with seeds from $0 - 9$ to obtain the mean and standard error.

For the figure of the decision boundaries (Figure 1c), we set $d_\text{inv} = d_\text{spu} = 1$ for easier visualization in 2D. We also set $\sigma_\text{inv}^2 = \sigma_\text{spu}^2 = 0.1$ and remove regularization, so that the plot is more perceptible than $\sigma_\text{inv}^2 = 0.25$. Nonetheless, we also show the decision boundaries in Figure 5 for $\sigma_\text{inv}^2 = \sigma_\text{spu}^2 = 0.25, d_\text{inv} = d_\text{spu} = 1, \lambda = 10^{-4}$, which essentially has the same pattern. The random seed is 0.

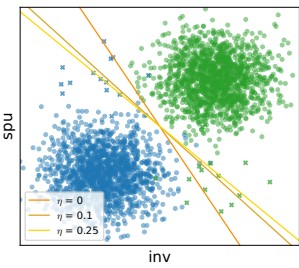

Figure 5: Decision Boundaries for $\sigma_{\text{inv}}^2 = \sigma_{\text{spu}}^2 = 0.25, d_{\text{inv}} = d_{\text{spu}} = 1, \lambda = 10^{-4}$

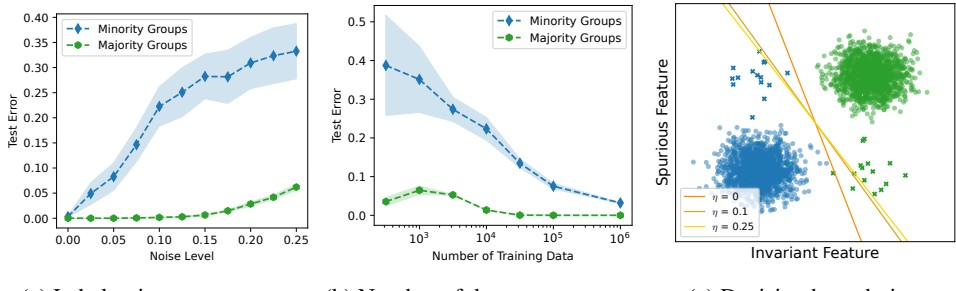

(a) Label noise vs. test error.    (b) Number of data vs. test error.    (c) Decision boundaries.

Figure 6: Simulations on *randomly projected* synthetic data trained with overparameterized logistic regression. The dotted lines are the means and the shaded regions are for the standard error from 5 independent runs. The result strongly resembles the case in Figure 1, showing that our result is also applicable to linearly transformed features.

## D.1 COMPARISON TO OTHER SYNTHETIC EXPERIMENTS

Sagawa et al. (2020) proposed the concept of spurious-core information ratio (SCR): The SCR is the ratio of the variance between the invariant (core) features and the spurious features: $\sigma_{\text{inv}}^2/\sigma_{\text{spu}}^2$. The higher the SCR, the more variations the invariant features. This will lead to noisier predictions for the invariant classifier (that only rely on the invariant features), subsequently resulting in the overparameterized models relying more on the spurious features. We do not assume that the SCR is high here and set $\sigma_{\text{inv}}^2 = \sigma_{\text{spu}}^2$, attaining a more general setting where invariant features are at least as informative as the spurious features in both noise-free and noisy settings.

## D.2 TRAINING WITH LINEARLY PROJECTED FEATURES

So far, the invariant features $\mathbf{x}_{\text{inv}}$ and spurious features $\mathbf{x}_{\text{spu}}$ occupy independent dimensions. In this section, we extend the synthetic setting such that invariant and spurious features are not perfectly disentangled into different dimensions. The only requirement is that they remain orthogonal to each other: $\mathbf{x}_{\text{inv}} \perp \mathbf{x}_{\text{spu}}$. To achieve this, we apply a random projection matrix to the dataset after sampling and rerun the same experiments. Based on the rotational invariance of logistic regression (Ng, 2004), we expect the results to not change by much. As shown in Figure 6, this is indeed the case, showing that our result is also applicable to linearly transformed features.

## E MAIN EXPERIMENTS

## E.1 DATASETS

All datasets are added to or implemented by the Domainbed (Gulrajani & Lopez-Paz, 2020) framework for better comparison among the algorithms.

### E.1.1 SUBPOPULATION SHIFTS WITH KNOWN GROUPS

**Waterbirds** (Sagawa et al., 2019) is a binary bird-type classification dataset adapted from CUB dataset (Wah et al., 2011) and Places dataset (Zhou et al., 2017). There are a total of four groups: waterbirds on water ($\mathcal{G}_1$), waterbirds on land ($\mathcal{G}_2$), landbirds on water ($\mathcal{G}_3$), and landbirds on land ($\mathcal{G}_4$). In the training set, there are 3498 (73%), 184 (4%), 56 (1%), and 1057 (22%) images for the 4 groups. Apparently, there is a strong spurious correlation between the bird type and the background in the training set. The majority groups are $\mathcal{G}_1$ and $\mathcal{G}_4$, and the minority groups are $\mathcal{G}_2, \mathcal{G}_3$. For the validation and test sets, there are an equal number of landbirds or waterbirds allocated to water and land backgrounds. We follow the standard train, val, and test splits.

**CelebA** (Sagawa et al., 2019) is a binary hair color blondness prediction dataset adapted from Liu et al. (2015). There are four groups: ($\mathcal{G}_1$) non-blond woman (71629 samples, 44%); ($\mathcal{G}_2$) non-blond man (66874 samples, 41%); ($\mathcal{G}_3$) blond woman (22880 samples, 14%); ($\mathcal{G}_4$) blond man (1387 samples, 1%). There exists a strong spurious correlation between gender man and non-blondness. The majority groups are $\mathcal{G}_1, \mathcal{G}_2, \mathcal{G}_3$, and the minority group is $\mathcal{G}_4$. We follow the standard train, val, and test splits.

**CivilComments** (Borkan et al., 2019) is a binary text toxicity classification task. We used the version implemented in WILDS (Koh et al., 2021). There are 16 overlapping groups based on 8 identities including male, female, LGBT, black, white, Christian, Muslim, other religion. In this work, we use the default setting and create 4 groups only based on the black identity.

### E.1.2 SUBPOPULATION SHIFTS WITHOUT GROUP INFORMATION

We create two new subpopulation-shift datasets adapted from real-world datasets including Waterbirds and CelebA:

**Waterbirds+**: Based on the 4 groups in Waterbirds, we create two new environments $e_1, e_2$. Each environment has half the data from each majority group ($\mathcal{G}_1, \mathcal{G}_4$). Then we allocate $1/3$ of the data from each minority group ($\mathcal{G}_2, \mathcal{G}_3$) to $e_1$ and the rest $2/3$ to $e_2$. This resembles the CMNIST setup so that $e_2$ has twice the amount of the minority data as $e_1$, such that background is not an invariant feature across the two environments because predicting with only the background features will result in different error rates across environments.

**CelebA+**: Similar to Waterbirds+, we create two new environments $e_1, e_2$. Each environment has half the data from each majority group ($\mathcal{G}_1, \mathcal{G}_2, \mathcal{G}_3$). Then we allocate $1/3$ of the data from the minority group ($\mathcal{G}_4$) to $e_1$ and the rest $2/3$ to $e_2$. Under this allocation, gender is not an invariant feature across the two environments because predicting hair blondness with only the gender features will result in different error rates across environments.

**CivilComments+**: Although there are 16 overlapping groups with 8 identities, we only create 4 groups based on the black identity. Thus, the setup can exactly follow Waterbirds+.

We also use the synthetic ColoredMNIST dataset:

**CMNIST** (Arjovsky et al., 2019) is one of the most commonly experimented synthetic domain generalization datasets. Based on MNIST, digits 0-4 and 5-9 are categorized into classes 0 and 1, respectively. Additional two-color information is added to the dataset by creating a separate channel, resulting in the input size being $(28 \times 28 \times 2)$. There are three environments $e \in \{0.1, 0.2, 0.9\}$ that mark the correlations between the color and the digits. In particular, in $e_1$, 10% of the green instances are 1 with the rest 90% being 0, and 10% of the red instances are 0 with the rest 90% being 1. Thus, using the color information to predict 1 with green and predict 0 with red, gives 90%, 80%, and 10% accuracy for the three environments. By default, the labels are corrupted with 25% uniform label noise.

### E.1.3 DOMAIN SHIFTS

We choose four domain-shift datasets from the Domainbed framework (Gulrajani & Lopez-Paz, 2020). All four datasets have input sizes (3, 224, 224).

**PACS** (Li et al., 2017) is a 7-class image classification dataset specifically designed for domain generalization, encompassing 4 distinct domains: Photo (1670 images), Art Painting (2048 images), Cartoon (2344 images), and Sketch (3929 images). There are 9991 samples in total.

**VLCS** (Fang et al., 2013) is a 5-class image dataset with 4 domains: Caltech101, LabelMe, SUN09, and VOC2007. Each domain is a standalone image classification dataset released by distinct research teams. There are 10729 samples in total.

**Office-Home** (Venkateswara et al., 2017) is a 65-class image classification dataset for objects that can be found typically in office or home environments. The dataset has 4 domains: Art, Clip Art, Product, and Real-World. There are 15588 samples in total.

**Terra Incognita** (Beery et al., 2018) is a 10-class wild-animal classification dataset with 4 domains: L100, L38, L43, L46. Each domain represents a distinct location where camera traps are placed for photography. There are 24788 samples in total.

## E.2 ALGORITHMS

We choose a few most influential and well-cited algorithms with theoretical guarantees for comparisons.

1. **Mixup** (Zhang et al., 2017) is a simple yet effective approach that has been widely studied from both empirical and theoretical perspectives. It interpolates the input and the loss of the output to create pseudo-training samples for better out-of-distribution generalization.

2. **IRM** (Arjovsky et al., 2019): IRM is the first algorithm proposed to learn invariant/causal predictors from multiple environments with deep learning. It augments the ERM with an additional objective of achieving optimal loss for all training environments simultaneously, preventing the model from learning spurious correlations that are only applicable to certain environments.

3. **V-REx** (Krueger et al., 2021): V-REx extends IRM with a simplified regularizer that minimizes the variance of risks across environments. It has the benefit of being robust to covariate shifts in the input distribution. It is also theoretically analyzed for homoskedastic environments (consistent noise level).

4. **GroupDRO** (Sagawa et al., 2019): GroupDRO prioritizes optimizing the worst-group risk to address subpopulation shifts. It is one of the first algorithms proposed to specifically address subpopulation shifts by minimizing the worst-group risk directly. By utilizing the group information effectively, it has been considered the gold standard amongst many subsequent works that try to deal with subpopulation shifts without group labels.

## E.3 EXPERIMENTAL SETUP

We use Domainbed (Gulrajani & Lopez-Paz, 2020) to run experiments for all 9 datasets. We report the performance (average and standard error) of the models trained out of 3 trials and each trial searches across 20 sets of different hyperparameters according to the default setting. We train all models for 5000 steps. for CMNIST, and 16 for the rest real-world datasets. We use a simple Convolutional Neural Network (CNN) for CMNIST, and ResNet 50 pretrained on Imagenet for the rest datasets. For all datasets, we add 10%, 25% label noise to test the performance of different algorithms.

For Waterbirds, CelebA, and their derivative Waterbirds+, CelebA+ datasets, we perform standard data augmentation according to Sagawa et al. (2019) using Pytorch (This is slightly different from the default augmentation implemented by Domainbed). Since Waterbirds and CelebA have predefined train, val, test splits and group information, we report both the average and worst-group (WG) test accuracy of the model selected according to the best WG performance in the standard validation split with early stopping, which follows most of the algorithms evaluated on these datasets.

For CMNIST, we do not perform any data augmentation. The evaluation is only done on the 3rd environment where the minority groups from the training environments become the majority. Due to the simplicity of the dataset, we have tested more fine-grained label noise $\eta \in \{0, 0.05, 0.1, 0.15, 0.2\}$.

For the domain-shift datasets, we adopt Domainbed's data augmentation (Gulrajani & Lopez-Paz, 2020). Since there are multiple different environments, we run a single-environment cross-test by taking the environments one by one as the test set and the rest as the training set. Then, we report the average accuracy of the models across all environments selected using 20% of the test data without early stopping.

# F    ADDITIONAL MAIN RESULTS

## F.1    CIVILCOMMENTS (NLP DATASET)

**CivilComments** (Borkan et al., 2019) is a binary text toxicity classification task. We used the version implemented in WILDS (Koh et al., 2021). There are 16 overlapping groups based on 8 identities including male, female, LGBT, black, white, Christian, Muslim, other religion. In this work, similar to Waterbirds and CelebA, we create 4 groups only based on black identity. We also simulate the two-environment subpopulation shift setting as done for Waterbirds+ and CelebA+, and we call it **CivilComments+**.

We use the base uncased DistillBERT model as the feature extractor. We feed the final-layer representation of the [CLS] token into a linear classifier for toxicity detection. We report the worst-group error with the worst-group selection. Note that Mixup cannot be directly applied to the NLP dataset since the input is discrete words that cannot be interpolated.

As shown in the Table below, GroupDRO has a clear advantage over other algorithms in the noise-free setting. However, ERM still performs competitively compared to other invariance learning algorithms such as IRM and V-REx, even under label noise setting, which is consistent with the existing empirical observations.

| Algorithm/$\eta$ | 0 | 0.1 | 0.25 | Avg |
|---|---|---|---|---|
| ERM | $89.8\pm_{0.4}$ | $67.9\pm_{0.4}$ | $55.4\pm_{0.1}$ | 71.0 |
| IRM | $89.2\pm_{0.2}$ | $66.9\pm_{0.4}$ | $54.2\pm_{0.3}$ | 70.1 |
| GroupDRO | $\mathbf{91.9}\pm_{0.2}$ | $\mathbf{69.7}\pm_{0.3}$ | $55.4\pm_{0.1}$ | **72.3** |
| VREx | $89.6\pm_{0.3}$ | $69.0\pm_{0.4}$ | $\mathbf{56.0}\pm_{0.3}$ | 71.6 |

Table 3: Worst-group accuracy (%) for subpopulation shift on CivilComments Dataset.

| Algorithm/$\eta$ | 0 | 0.1 | 0.25 | Avg |
|---|---|---|---|---|
| ERM | $69.2\pm_{1.2}$ | $59.8\pm_{1.9}$ | $52.5\pm_{1.0}$ | 60.5 |
| IRM | $60.3\pm_{4.5}$ | $54.8\pm_{4.0}$ | $50.3\pm_{1.8}$ | 55.1 |
| GroupDRO | $\mathbf{73.3}\pm_{0.4}$ | $\mathbf{60.2}\pm_{0.9}$ | $52.2\pm_{0.5}$ | **61.9** |
| VREx | $64.5\pm_{4.2}$ | $58.7\pm_{1.7}$ | $\mathbf{54.9}\pm_{0.3}$ | 59.4 |

Table 4: Worst-group accuracy (%) for subpopulation shift on CivilComments+ Dataset.

## F.2    EVALUATING MORE ALGORITHMS

We added evaluations of four additional algorithms used for domain adaptation and domain generalization, including:

1. **CORAL** (Sun & Saenko, 2016): A moment matching technique that matches the mean and variance of the features between the training and test domains.
2. **IB_IRM** (Ahuja et al., 2021): An IRM based algorithm that is combined with information bottleneck as an additional regularization.
3. **Fish** (Shi et al., 2021): A gradient matching technique that aligns the gradients of different training domains by maximizing their inner products.
4. **Fishr** (Rame et al., 2022): A gradient matching technique based on regularizing the inter-domain variance of the intra-domain gradient variance.

We show the results for three datasets from each category, including corrupted CMNIST (synthetic subpopulation shift), Waterbirds+ (real-world subpopulation shift), and PACS (real-world domain shift). We observe that similar trend still holds compared to other DG algorithms presented in the main paper. Thus, we hypothesize that these results can be extended to other datasets that are similar to the ones evaluated.

### F.2.1 CMNIST

| Algorithm/$\eta$ | 0 | 0.1 | 0.25 | Avg |
|---|---|---|---|---|
| ERM | $97.1\pm_{0.1}$ | $77.8\pm_{1.9}$ | $39.5\pm_{1.2}$ | 71.5 |
| Mixup | $\mathbf{97.3}\pm_{0.2}$ | $82.3\pm_{0.9}$ | $24.7\pm_{1.9}$ | 68.1 |
| GroupDRO | $\mathbf{97.3}\pm_{0.1}$ | $84.5\pm_{1.7}$ | $48.3\pm_{1.0}$ | 76.7 |
| IRM | $96.8\pm_{0.3}$ | $\mathbf{86.5}\pm_{3.8}$ | $\mathbf{73.2}\pm_{9.1}$ | $\mathbf{85.5}$ |
| VREx | $96.7\pm_{0.1}$ | $84.1\pm_{1.8}$ | $57.8\pm_{2.0}$ | 79.5 |
| CORAL | $97.1\pm_{0.2}$ | $77.3\pm_{1.4}$ | $37.2\pm_{0.7}$ | 70.5 |
| IB_IRM | $96.5\pm_{0.5}$ | $83.7\pm_{4.4}$ | $71.3\pm_{5.8}$ | 83.8 |
| Fish | $97.2\pm_{0.2}$ | $78.8\pm_{0.6}$ | $37.7\pm_{1.3}$ | 71.2 |
| Fishr | $\mathbf{97.3}\pm_{0.0}$ | $82.0\pm_{1.3}$ | $67.6\pm_{9.4}$ | 82.3 |

### F.2.2 WATERBIRDS+

| Algorithm/$\eta$ | 0 | 0.1 | 0.25 | Avg |
|---|---|---|---|---|
| ERM | $81.5\pm_{0.4}$ | $64.6\pm_{0.5}$ | $50.4\pm_{0.9}$ | 65.5 |
| IRM | $78.5\pm_{0.8}$ | $63.3\pm_{0.5}$ | $47.8\pm_{1.8}$ | 63.2 |
| GroupDRO | $\mathbf{82.6}\pm_{0.4}$ | $65.6\pm_{0.8}$ | $51.2\pm_{0.7}$ | $\mathbf{66.5}$ |
| Mixup | $79.8\pm_{0.9}$ | $61.8\pm_{2.0}$ | $51.3\pm_{1.5}$ | 64.3 |
| VREx | $78.3\pm_{0.3}$ | $\mathbf{66.6}\pm_{0.9}$ | $51.9\pm_{0.5}$ | 65.6 |
| CORAL | $80.8\pm_{0.6}$ | $64.3\pm_{1.8}$ | $\mathbf{53.5}\pm_{1.8}$ | 66.2 |
| IB_IRM | $74.5\pm_{1.2}$ | $60.8\pm_{1.5}$ | $48.6\pm_{1.4}$ | 61.3 |
| Fish | $82.3\pm_{0.3}$ | $66.4\pm_{0.4}$ | $49.8\pm_{2.4}$ | 66.2 |
| Fishr | $79.4\pm_{0.7}$ | $62.1\pm_{0.5}$ | $\mathbf{53.5}\pm_{1.3}$ | 65.0 |

### F.2.3 PACS

| Algorithm/$\eta$ | 0.1 | 0.25 | Avg |
|---|---|---|---|
| ERM | $82.0\pm_{0.5}$ | $74.5\pm_{0.6}$ | 78.3 |
| IRM | $80.4\pm_{1.2}$ | $71.0\pm_{1.9}$ | 75.7 |
| GroupDRO | $82.4\pm_{0.3}$ | $74.7\pm_{0.4}$ | 78.5 |
| Mixup | $\mathbf{83.6}\pm_{0.1}$ | $75.2\pm_{0.9}$ | $\mathbf{79.4}$ |
| VREx | $81.4\pm_{0.2}$ | $73.5\pm_{0.7}$ | 77.4 |
| CORAL | $82.6\pm_{0.4}$ | $75.9\pm_{0.3}$ | 79.2 |
| IB_IRM | $77.0\pm_{2.5}$ | $67.8\pm_{3.5}$ | 72.4 |
| Fish | $83.1\pm_{0.1}$ | $\mathbf{77.0}\pm_{0.7}$ | $\mathbf{80.1}$ |
| Fishr | $81.7\pm_{0.4}$ | $74.5\pm_{0.5}$ | 78.1 |

## G LIMITATIONS

### G.1 FEATURE AVAILABILITY

In practice, when the feature extractors for image and text are commonly non-linear, it is uncertain whether the invariant and spurious features can be perfectly extracted by the deep learning model. As our setup assumes the availability of invariant and spurious features, our theoretical results in Section 4.1 under linear settings may not be directly applicable in the real world.

We would like to discuss two aspects regarding the potential limitations due to the availability of invariant, spurious, and nuisance features:

1. **Failure mode of ERM:** Let's briefly consider two other scenarios where the ideal feature extractor does not exist:

    (a) Spurious features are not fully extracted: This can be viewed as having a weaker spurious correlation and our results still apply.

    (b) Invariant features are not fully extracted: The model has to exploit more spurious correlations and nuisance features, so it already fails to generalize.

As such, we believe that showing the failure mode of ERM under such an ideal situation actually encompasses some other failure modes without the ideal feature extractors. The assumption of having orthogonal and spurious features is not restrictive to demonstrate the failure mode of ERM.

2. **Advantage of DG algorithms in practice:** Typically, DG algorithms aim to learn the invariant representations and require the existence of invariant features to function. If the invariant features cannot be fully extracted, there is also no guarantee for DG algorithms to have the advantage over ERM. The robustness to label noise does not necessarily imply that the invariant representations could be learned. Rather, it has the implication of less memorization and not converging to the suboptimal solution when label noise is present. Thus, the unavailability of the ideal sets of features could be responsible for DG algorithms not having a clear advantage over ERM in real-world noisy datasets. However, without enough evidence, we do think that pretraining the model with better representations is more responsible for the reduced performance gaps between different DG algorithms vs. ERM compared to the lack of an ideal feature extractor.

## G.2 Linear Separability

Our theoretical analysis is also based on the assumption that the invariant features are linearly separable in the noise-free setting. However, in practice, the invariant features can achieve the Bayes optimal error rate $> 0\%$, making the invariant features non-linearly separable. Our theorem then is not directly applicable. Future work that explicitly allows non-separable invariant features will improve the analysis.

However, we think that our assumption is in fact more inclusive than it is restrictive. This is because the linear separability is only assumed for the noise-free data distribution. In the cases when the dataset is not linearly separable, we can view that as a result of adding some form of label noise to the clean data. A higher noise level corresponds to worse non-separability. However, we acknowledge that this analogy is by no means comprehensive of all cases.

## G.3 Generalization to Other DG Methods

Our analysis in Section 5 addresses the noise robustness of two invariance learning algorithms: IRM and V-REx. However, as DG algorithms are designed for various aspects of the training regime with diverse mechanisms (discussed in Section 2), not all domain generalization methods can be theoretically proven to possess such a desirable property. For example, from the empirical observation in Appendix F.2, algorithms such as Fish which aligns the gradient of samples between environments may not be robust to noise on the CMNIST dataset. More comprehensive analyses are needed to incorporate all algorithms.

## H  Noise Memorization on Waterbirds Data

| Algorithm | Waterbirds+ 0.1 | Waterbirds+ 0.25 | Waterbirds 0.1 | Waterbirds 0.25 |
|---|---|---|---|---|
| ERM | $28.3\pm_{3.8}$ | $35.1\pm_{0.7}$ | $33.4\pm_{0.7}$ | $58.4\pm_{5.6}$ |
| IRM | $27.3\pm_{5.6}$ | $38.7\pm_{1.8}$ | $30.4\pm_{0.9}$ | $54.3\pm_{4.7}$ |
| GroupDRO | $23.4\pm_{2.1}$ | $36.3\pm_{0.6}$ | $32.9\pm_{0.1}$ | $57.3\pm_{3.7}$ |
| Mixup | $28.2\pm_{3.6}$ | $34.6\pm_{0.6}$ | $34.9\pm_{0.9}$ | $57.2\pm_{3.0}$ |
| VREx | $24.8\pm_{4.9}$ | $37.8\pm_{0.9}$ | $35.3\pm_{0.9}$ | $48.1\pm_{0.3}$ |

Table 5: Training noise memorization on Waterbirds+

As shown in Table 5, the memorization issue is much less severe. The likely reason is because early-stopping is applied to Waterbirds, CelebA, and CivilComments, so that model checkpoints in the early-training steps with less memorization are selected. Besides, pretraining on ImageNet allows the models to fit the downstream tasks faster. Thus, good performance can be obtained without prolonged training compared to training from scratch.

## I  IMPLEMENTATION DETAILS OF ERM

The classic environment-agnostic ERM randomly samples training data from a pool of data from all environments. For the environment-balanced ERM algorithm implemented by Domainbed (Gulrajani & Lopez-Paz, 2020), in each step, an equal amount of data is sampled from each environment according to batch_size. Thus, the actual "batch size" equals to num_data × batch_size. Datasets with more environments are trained effectively with more data points.

## J  RESULTS

All these results are obtained using 5000 training steps and dataset-dependent batch size.

### J.1  CMNIST

#### J.1.1  AVERAGES

| Algorithm/$\eta$ | 0 | 0.05 | 0.1 | 0.15 | 0.25 | Avg |
|---|---|---|---|---|---|---|
| ERM | $97.1\pm_{0.1}$ | $89.2\pm_{0.6}$ | $77.8\pm_{1.9}$ | $63.1\pm_{0.8}$ | $39.5\pm_{1.2}$ | 73.3 |
| IRM | $96.8\pm_{0.3}$ | $89.5\pm_{1.1}$ | $\mathbf{86.5}\pm_{3.8}$ | $76.6\pm_{7.0}$ | $\mathbf{73.2}\pm_{9.1}$ | $\mathbf{84.5}$ |
| GroupDRO | $\mathbf{97.3}\pm_{0.1}$ | $\mathbf{91.1}\pm_{0.7}$ | $84.5\pm_{1.7}$ | $74.9\pm_{1.7}$ | $48.3\pm_{1.0}$ | 79.2 |
| Mixup | $\mathbf{97.3}\pm_{0.2}$ | $90.8\pm_{1.1}$ | $82.3\pm_{0.9}$ | $66.4\pm_{2.9}$ | $24.7\pm_{1.9}$ | 72.3 |
| VREx | $96.7\pm_{0.1}$ | $90.9\pm_{0.8}$ | $84.1\pm_{1.8}$ | $\mathbf{82.3}\pm_{3.1}$ | $57.8\pm_{2.0}$ | 82.3 |

#### J.1.2  NOISE MEMORIZATION

| Algorithm/$\eta$ | 0.05 | 0.1 | 0.15 | 0.25 | Avg |
|---|---|---|---|---|---|
| ERM | $26.6\pm_{1.5}$ | $31.7\pm_{6.4}$ | $53.7\pm_{3.7}$ | $74.5\pm_{1.6}$ | 46.6 |
| IRM | $16.6\pm_{3.6}$ | $16.6\pm_{5.8}$ | $32.0\pm_{12.2}$ | $\mathbf{26.8}\pm_{9.0}$ | 23.0 |
| GroupDRO | $10.0\pm_{1.2}$ | $\mathbf{15.7}\pm_{1.7}$ | $26.2\pm_{2.2}$ | $54.5\pm_{2.8}$ | 26.6 |
| Mixup | $17.5\pm_{1.0}$ | $22.1\pm_{1.1}$ | $38.1\pm_{3.6}$ | $77.7\pm_{1.9}$ | 38.9 |
| VREx | $\mathbf{8.3}\pm_{0.8}$ | $16.6\pm_{2.0}$ | $\mathbf{18.3}\pm_{4.0}$ | $41.2\pm_{2.2}$ | $\mathbf{21.1}$ |

### J.2  WATERBIRDS

#### J.2.1  WORST-GROUP TEST ACCURACY

| Algorithm | 0 | 0.1 | 0.25 | Avg |
|---|---|---|---|---|
| ERM | $86.7\pm_{1.0}$ | $61.4\pm_{1.3}$ | $48.8\pm_{1.2}$ | $\mathbf{65.6}$ |
| IRM | $87.0\pm_{1.4}$ | $58.8\pm_{0.8}$ | $48.3\pm_{0.8}$ | 64.7 |
| GroupDRO | $84.9\pm_{0.7}$ | $\mathbf{61.9}\pm_{0.7}$ | $47.7\pm_{1.3}$ | 64.8 |
| Mixup | $\mathbf{88.0}\pm_{0.4}$ | $56.9\pm_{0.9}$ | $\mathbf{50.4}\pm_{0.1}$ | 65.1 |
| VREx | $87.4\pm_{0.6}$ | $59.2\pm_{1.5}$ | $46.8\pm_{0.4}$ | 64.5 |

#### J.2.2  AVERAGE TEST ACCURACY

| Algorithm | 0 | 0.1 | 0.25 | Avg |
|---|---|---|---|---|
| ERM | $91.9\pm_{0.4}$ | $63.6\pm_{0.6}$ | $50.3\pm_{0.9}$ | 68.6 |
| IRM | $91.8\pm_{0.4}$ | $63.5\pm_{1.0}$ | $50.6\pm_{0.4}$ | 68.6 |
| GroupDRO | $91.9\pm_{0.3}$ | $63.6\pm_{0.8}$ | $51.1\pm_{0.2}$ | 68.9 |
| Mixup | $92.6\pm_{0.3}$ | $62.2\pm_{0.8}$ | $51.2\pm_{0.1}$ | 68.7 |
| VREx | $92.6\pm_{0.7}$ | $63.7\pm_{1.4}$ | $51.0\pm_{0.2}$ | 69.1 |

## J.3 WATERBIRDS+

### J.3.1 WORST-GROUP TEST ACCURACY

| Algorithm | 0 | 0.1 | 0.25 | Avg |
|-----------|---|-----|------|-----|
| ERM | $81.5 \pm_{0.4}$ | $64.6 \pm_{0.5}$ | $50.4 \pm_{0.9}$ | 65.5 |
| IRM | $78.5 \pm_{0.8}$ | $63.3 \pm_{0.5}$ | $47.8 \pm_{1.8}$ | 63.2 |
| GroupDRO | $\mathbf{82.6} \pm_{0.4}$ | $65.6 \pm_{0.8}$ | $51.2 \pm_{0.7}$ | **66.5** |
| Mixup | $79.8 \pm_{0.9}$ | $61.8 \pm_{2.0}$ | $51.3 \pm_{1.5}$ | 64.3 |
| VREx | $78.3 \pm_{0.3}$ | $\mathbf{66.6} \pm_{0.9}$ | $\mathbf{51.9} \pm_{0.5}$ | 65.6 |

### J.3.2 AVERAGE TEST ACCURACY

| Algorithm | 0 | 0.1 | 0.25 | Avg |
|-----------|---|-----|------|-----|
| ERM | $90.3 \pm_{0.1}$ | $\mathbf{73.5} \pm_{0.4}$ | $55.7 \pm_{0.2}$ | **73.2** |
| IRM | $89.0 \pm_{0.8}$ | $72.0 \pm_{1.3}$ | $\mathbf{56.4} \pm_{0.3}$ | 72.5 |
| GroupDRO | $\mathbf{90.4} \pm_{0.2}$ | $70.6 \pm_{1.1}$ | $54.3 \pm_{1.3}$ | 71.8 |
| Mixup | $89.7 \pm_{0.5}$ | $72.2 \pm_{1.6}$ | $54.7 \pm_{0.8}$ | 72.2 |
| VREx | $89.6 \pm_{0.7}$ | $71.6 \pm_{1.2}$ | $55.2 \pm_{0.5}$ | 72.1 |

## J.4 CELEBA

### J.4.1 WORST-GROUP TEST ACCURACY

| Algorithm | CelebA | CelebA 0.1 | CelebA 0.25 | Avg |
|-----------|--------|------------|-------------|-----|
| ERM | $88.1 \pm_{1.2}$ | $\mathbf{63.1} \pm_{1.1}$ | $50.2 \pm_{1.2}$ | **67.1** |
| IRM | $86.9 \pm_{0.5}$ | $61.7 \pm_{1.8}$ | $\mathbf{51.2} \pm_{0.5}$ | 66.6 |
| GroupDRO | $88.0 \pm_{0.4}$ | $59.3 \pm_{1.7}$ | $49.0 \pm_{1.4}$ | 65.4 |
| Mixup | $87.6 \pm_{0.4}$ | $63.0 \pm_{1.4}$ | $49.5 \pm_{0.5}$ | 66.7 |
| VREx | $\mathbf{89.3} \pm_{0.7}$ | $60.2 \pm_{2.0}$ | $49.5 \pm_{1.4}$ | 66.3 |

### J.4.2 AVERAGE TEST ACCURACY

| Algorithm | 0 | 0.1 | 0.25 | Avg |
|-----------|---|-----|------|-----|
| ERM | $93.5 \pm_{0.2}$ | $67.2 \pm_{0.1}$ | $52.3 \pm_{0.1}$ | 71.0 |
| IRM | $93.5 \pm_{0.1}$ | $67.1 \pm_{0.2}$ | $52.1 \pm_{0.1}$ | 70.9 |
| GroupDRO | $\mathbf{93.8} \pm_{0.1}$ | $67.0 \pm_{0.2}$ | $52.1 \pm_{0.0}$ | 70.9 |
| Mixup | $93.5 \pm_{0.1}$ | $67.6 \pm_{0.2}$ | $\mathbf{52.6} \pm_{0.1}$ | **71.2** |
| VREx | $93.5 \pm_{0.1}$ | $\mathbf{67.8} \pm_{0.1}$ | $52.5 \pm_{0.1}$ | **71.2** |

## J.5 CELEBA+

### J.5.1 WORST-GROUP TEST ACCURACY

| Algorithm | 0 | 0.1 | 0.25 | Avg |
|-----------|---|-----|------|-----|
| ERM | $71.7 \pm_{2.9}$ | $65.4 \pm_{1.9}$ | $55.0 \pm_{1.7}$ | 64.0 |
| IRM | $\mathbf{75.4} \pm_{4.3}$ | $62.4 \pm_{2.5}$ | $53.0 \pm_{1.2}$ | 63.6 |
| GroupDRO | $71.9 \pm_{2.0}$ | $61.7 \pm_{1.7}$ | $\mathbf{57.0} \pm_{1.7}$ | 63.5 |
| Mixup | $71.1 \pm_{1.2}$ | $64.6 \pm_{1.8}$ | $52.6 \pm_{1.7}$ | 62.8 |
| VREx | $73.9 \pm_{0.7}$ | $\mathbf{69.9} \pm_{2.0}$ | $50.0 \pm_{0.7}$ | **64.6** |

### J.5.2 Average Test Accuracy

| Algorithm | CelebA | CelebA 0.1 | CelebA 0.25 | Avg |
|-----------|--------|------------|-------------|-----|
| ERM | $91.9\pm_{1.5}$ | $76.2\pm_{1.1}$ | $60.3\pm_{0.4}$ | 76.1 |
| IRM | $89.6\pm_{1.0}$ | $76.5\pm_{0.5}$ | $59.8\pm_{0.7}$ | 75.3 |
| GroupDRO | $89.9\pm_{1.6}$ | $76.9\pm_{0.2}$ | $60.5\pm_{0.1}$ | 75.8 |
| Mixup | $91.6\pm_{1.3}$ | $76.6\pm_{0.2}$ | $60.2\pm_{0.5}$ | 76.1 |
| VREx | $90.5\pm_{0.2}$ | $74.8\pm_{0.2}$ | $60.3\pm_{0.3}$ | 75.2 |
| Fishr | $92.3\pm_{0.9}$ | $75.8\pm_{0.4}$ | $60.7\pm_{0.3}$ | 76.3 |

### J.6 PACS

#### J.6.1 PACS $\eta$=0.1

| Algorithm | A | C | P | S | Avg |
|-----------|---|---|---|---|-----|
| ERM | $81.1\pm_{0.4}$ | $76.4\pm_{0.1}$ | $95.1\pm_{0.3}$ | $75.3\pm_{1.7}$ | 82.0 |
| IRM | $77.0\pm_{0.9}$ | $75.7\pm_{2.1}$ | $96.0\pm_{0.3}$ | $72.7\pm_{2.3}$ | 80.4 |
| GroupDRO | $81.7\pm_{0.8}$ | $78.0\pm_{0.8}$ | $94.0\pm_{0.3}$ | $75.8\pm_{0.7}$ | 82.4 |
| Mixup | $84.2\pm_{0.6}$ | $77.8\pm_{0.2}$ | $96.3\pm_{0.4}$ | $76.0\pm_{0.1}$ | 83.6 |
| VREx | $80.3\pm_{0.3}$ | $75.9\pm_{1.0}$ | $95.0\pm_{0.4}$ | $74.4\pm_{1.4}$ | 81.4 |

#### J.6.2 PACS $\eta$=0.25

| Algorithm | A | C | P | S | Avg |
|-----------|---|---|---|---|-----|
| ERM | $73.6\pm_{0.4}$ | $67.9\pm_{0.2}$ | $87.4\pm_{1.0}$ | $69.3\pm_{1.5}$ | 74.5 |
| IRM | $72.8\pm_{0.4}$ | $64.9\pm_{3.7}$ | $90.0\pm_{0.9}$ | $56.4\pm_{4.1}$ | 71.0 |
| GroupDRO | $74.4\pm_{1.0}$ | $67.7\pm_{1.6}$ | $89.9\pm_{0.8}$ | $66.9\pm_{0.8}$ | 74.7 |
| Mixup | $72.4\pm_{1.2}$ | $69.9\pm_{0.6}$ | $90.5\pm_{1.0}$ | $67.8\pm_{2.2}$ | 75.2 |
| VREx | $69.8\pm_{1.2}$ | $68.4\pm_{0.5}$ | $89.4\pm_{0.9}$ | $66.2\pm_{1.0}$ | 73.5 |

### J.7 VLCS

#### J.7.1 VLCS $\eta$=0.1

| Algorithm | C | L | S | V | Avg |
|-----------|---|---|---|---|-----|
| ERM | $97.2\pm_{0.3}$ | $64.4\pm_{0.5}$ | $68.8\pm_{1.0}$ | $69.8\pm_{0.7}$ | 75.0 |
| IRM | $96.7\pm_{0.7}$ | $64.0\pm_{1.0}$ | $67.8\pm_{0.3}$ | $69.9\pm_{0.7}$ | 74.6 |
| GroupDRO | $96.1\pm_{0.9}$ | $64.6\pm_{0.2}$ | $69.2\pm_{0.8}$ | $70.6\pm_{1.4}$ | 75.1 |
| Mixup | $97.1\pm_{0.2}$ | $65.5\pm_{1.1}$ | $69.2\pm_{0.3}$ | $70.1\pm_{0.4}$ | 75.5 |
| VREx | $95.1\pm_{1.0}$ | $65.9\pm_{1.1}$ | $68.2\pm_{0.4}$ | $70.6\pm_{0.3}$ | 75.0 |

#### J.7.2 VLCS $\eta$=0.25

| Algorithm | C | L | S | V | Avg |
|-----------|---|---|---|---|-----|
| ERM | $93.6\pm_{0.8}$ | $62.7\pm_{1.3}$ | $64.6\pm_{0.9}$ | $66.8\pm_{0.9}$ | 71.9 |
| IRM | $94.5\pm_{0.4}$ | $62.1\pm_{0.3}$ | $61.6\pm_{0.6}$ | $63.1\pm_{1.0}$ | 70.3 |
| GroupDRO | $93.6\pm_{1.1}$ | $62.4\pm_{0.4}$ | $63.8\pm_{0.8}$ | $65.1\pm_{1.1}$ | 71.2 |
| Mixup | $94.1\pm_{0.3}$ | $62.5\pm_{0.8}$ | $64.0\pm_{0.3}$ | $66.9\pm_{1.2}$ | 71.9 |
| VREx | $93.9\pm_{1.0}$ | $63.0\pm_{0.6}$ | $63.6\pm_{1.5}$ | $66.8\pm_{0.6}$ | 71.8 |

## J.8 OFFICEHOME

### J.8.1 OFFICEHOME $\eta$=0.1

| Algorithm | A | C | P | R | Avg |
|---|---|---|---|---|---|
| ERM | $57.4\pm_{0.6}$ | $47.4\pm_{0.4}$ | $71.1\pm_{0.3}$ | $72.7\pm_{0.3}$ | 62.2 |
| IRM | $55.6\pm_{1.1}$ | $46.3\pm_{1.6}$ | $70.7\pm_{1.1}$ | $72.1\pm_{1.0}$ | 61.2 |
| GroupDRO | $55.8\pm_{1.0}$ | $47.0\pm_{0.3}$ | $69.5\pm_{1.0}$ | $72.8\pm_{0.4}$ | 61.3 |
| Mixup | $57.0\pm_{0.3}$ | $51.8\pm_{0.5}$ | $73.2\pm_{0.4}$ | $73.8\pm_{0.5}$ | 63.9 |
| VREx | $54.7\pm_{1.2}$ | $47.6\pm_{0.3}$ | $68.9\pm_{0.7}$ | $71.1\pm_{0.1}$ | 60.6 |

### J.8.2 OFFICEHOME $\eta$=0.25

| Algorithm | A | C | P | R | Avg |
|---|---|---|---|---|---|
| ERM | $47.9\pm_{1.1}$ | $41.3\pm_{1.4}$ | $64.3\pm_{0.4}$ | $66.0\pm_{0.7}$ | 54.9 |
| IRM | $46.9\pm_{1.9}$ | $39.1\pm_{2.4}$ | $63.2\pm_{1.4}$ | $66.5\pm_{1.9}$ | 53.9 |
| GroupDRO | $48.6\pm_{0.4}$ | $39.2\pm_{0.7}$ | $62.4\pm_{0.6}$ | $66.1\pm_{0.4}$ | 54.1 |
| Mixup | $53.1\pm_{0.6}$ | $42.0\pm_{1.2}$ | $66.3\pm_{0.2}$ | $68.4\pm_{0.1}$ | 57.5 |
| VREx | $47.1\pm_{2.4}$ | $39.1\pm_{0.8}$ | $61.0\pm_{0.3}$ | $64.6\pm_{0.8}$ | 53.0 |

## J.9 TERRAINCOGNITA

### J.9.1 TERRAINCOGNITA $\eta$=0.1

| Algorithm | L100 | L38 | L43 | L46 | Avg |
|---|---|---|---|---|---|
| ERM | $58.1\pm_{1.9}$ | $52.1\pm_{1.0}$ | $57.9\pm_{0.4}$ | $40.5\pm_{0.4}$ | 52.1 |
| IRM | $47.6\pm_{3.3}$ | $51.9\pm_{1.2}$ | $53.4\pm_{1.0}$ | $40.4\pm_{2.5}$ | 48.3 |
| GroupDRO | $58.0\pm_{1.5}$ | $49.1\pm_{1.0}$ | $59.3\pm_{0.2}$ | $40.8\pm_{0.5}$ | 51.8 |
| Mixup | $62.4\pm_{1.7}$ | $51.4\pm_{2.0}$ | $58.3\pm_{1.0}$ | $40.6\pm_{1.2}$ | 53.2 |
| VREx | $49.7\pm_{0.9}$ | $46.7\pm_{0.9}$ | $56.0\pm_{1.2}$ | $40.8\pm_{0.6}$ | 48.3 |

### J.9.2 TERRAINCOGNITA $\eta$=0.25

| Algorithm | L100 | L38 | L43 | L46 | Avg |
|---|---|---|---|---|---|
| ERM | $52.3\pm_{0.2}$ | $48.8\pm_{0.3}$ | $55.8\pm_{0.6}$ | $37.2\pm_{1.4}$ | 48.5 |
| IRM | $41.5\pm_{5.9}$ | $46.1\pm_{0.2}$ | $51.0\pm_{2.3}$ | $39.0\pm_{2.3}$ | 44.4 |
| GroupDRO | $56.9\pm_{2.7}$ | $48.7\pm_{0.4}$ | $55.4\pm_{1.0}$ | $41.5\pm_{0.6}$ | 50.6 |
| Mixup | $62.2\pm_{1.8}$ | $47.5\pm_{1.4}$ | $55.3\pm_{0.6}$ | $40.2\pm_{1.0}$ | 51.3 |
| VREx | $51.0\pm_{2.6}$ | $51.5\pm_{1.2}$ | $54.6\pm_{0.5}$ | $38.5\pm_{0.9}$ | 48.9 |

