# OpenReview forum: "Understanding Domain Generalization: A Noise Robustness Perspective"
_ICLR.cc/2024/Conference — ICLR 2024 poster_

### Official Review · Reviewer_qd9a · 2023-10-28

**Soundness:** 2 fair
**Presentation:** 2 fair
**Contribution:** 2 fair
**Rating:** 5
**Confidence:** 5

**Summary:**

This paper theoretically analyzes the benefits of the DG algorithm over ERM from the perspective of noise robustness. Label noise increases the model’s dependence on the spurious features of ERM. In contrast, DG algorithms have better label-noise robustness with the regularization which prevents capturing spurious correlation.

**Strengths:**

1. The authors theoretically analyze the benefits of the DG algorithm over ERM from the perspective of noise robustness.
2. Experimental results show that DG algorithms are more robust to label noise memorization.
3. The authors provide discussions about why noise robustness couldn’t lead to better performance of the DG algorithm in practical scenarios.

**Weaknesses:**

1.	My primary concern relates to the assumption of linear separability, as it can be challenging to meet this condition in real-world settings, especially when the invariant features are sparse. Moreover, it is also hard to have orthogonal invariant and spurious features. I think a more general analysis should be given to support the main idea.

2.	In practical scenarios, the minimum-norm classifier may not be the most suitable choice, which implies that Thm4.2 may not be applicable.

3.	The theorem does not sufficiently prove the ideas presented in this paper. Thm4.2 only suggests that the ERM algorithm favors spurious features over invariant ones, which leads to poor generalization performance.  However, for the DG algorithm, the authors only incorporate a regularization term and analyze the gradient with varying lambda. No formal theorem is there to demonstrate the superior performance of the DG algorithm.

4.	The crucial information is not expressed clearly. In page 4, how comes "the classifier become either $w_spu$ or $w_inv$"?  Is it an assumption or is it a mathematically grounded result? I think the authors should justify such an extreme claim.

5.	The description of the Lemma C.1 is unclear. The Lemma measures the cost of memorizing a mislabeled or non-spuriously correlated sample. Based on the derivation process, it should be a conclusion that holds with high probability, but the lemma presents it deterministically.

6.	The absence of mathematical symbol definitions in this paper reduces its readability. Some of the symbols are not described (just mentioned) in this paper. If possible, give their mathematical definitions.

7.	The experiments do not effectively support the theorem. In the experiment section, I recommend that the authors discuss the values of the weights' norms, such as $w^(inv)_inv$, $w^(spu)_spu$, $w^(inv)$, and $w^(spu)$, to further support the proof of Thm4.2.

**Questions:**

1.	In all tables in experiments, the authors should add references to Mixeup, GroupDRO, IRM, VREx.
2.	The author should provide another theorem to illustrate that the classifier with a smaller norm becomes more favored by the model.
3.	The authors should provide further explanations on the regularization term in the practical surrogate. This should include an explanation of why it has this specific form and why it incorporates first-order derivatives.

---

> ### Author Response · Authors · 2023-11-18
> **Author Response to Reviewer qd9a (Part 1)**
>
> We would like to thank you for appreciating our analyses of the harmful effect of label noise with spurious correlations and the noise robustness of DG algorithms. We address your comments below.
>
> **Summary:**
> - We provide an extensive discussion to your concern about our assumptions and the applicability of the theorems.
> - We provide theoretical justifications for the implicit bias towards minimum-norm solution and an additional empirical study to the norms of the classifiers.
> - We thank your valuable suggestions for improving our paper, which has been revised accordingly in blue color.
>
> **Weaknesses:**
>
> 1. > My primary concern relates to the assumption of linear separability ...
>
>    We think that the assumption is in fact more inclusive than it is restrictive, as discussed right after Assumption 4.1. This is because the linear separability is only assumed for the noise-free data distribution. In cases when the dataset is not linearly separable, we can view that as a result of adding some form of label noise to the clean data. For example, when some critical features are not observed, there may exist the same data points with different labels, and some can be viewed as label noise. A higher noise level corresponds to worse non-separability. However, we acknowledge that this analogy is by no means comprehensive of all cases.
>
>    > Moreover, it is also hard to have orthogonal invariant and spurious features...
>
>    We believe that the orthogonality is actually a reasonable setup. Recall that by definition the invariant features are optimal across all environments. An alternative interpretation is that these features do not change even though the environment changes. For example, when classifying the cow/camel, the features of the foreground animal should not change as the background gets changed to grassland or desert. Now consider the case where invariant and spurious features are not orthogonal, then the spurious features will have a non-zero projection onto the invariant features. As the spurious features change across environments, its projection onto the invariant features also changes. This causes the invariant features to change, which violates the definition of invariance. Thus, it is common to assume a feature vector can be decomposed into spurious and invariant components that are orthogonal to each other. A similar assumption has also been adopted in prior works [3,4].
>
> 2. > In practical scenarios, the minimum-norm classifier may not be the most suitable choice, which implies that Thm4.2 may not be applicable.
>
>    For linear cases, we add two formal theorems shown by [5,6] related to the implicit regularization effect of gradient descent (GD) in Appendix C.2. In particular, [5] proved that logistic regression converges in the direction of the max-margin classifier, which is equivalent to the min-norm classifier given the right constraints. Thus, we can conclude that the model has the inductive bias of learning models with smaller norms. In practice, when the model becomes a neural network, and early stopping is often used, it is true that the min-norm may not always be favored. On the bright side, homogeneous neural networks have been shown to converge in direction to max-margin solution [2] as well. There is a potential to extend the current analysis and we leave that for future work.
>
> 3. > The theorem does not sufficiently prove the ideas presented in this paper. ... No formal theorem is there to demonstrate the superior performance of the DG algorithm.
>
>    Our norm-related analysis in Section 4.2 is independent of the algorithms, but the theorem is only applicable to ERM because it has been shown to have the implicit regularization effect with GD. For IRM and other DG algorithms, there is no guarantee that gradient descent still favors the small norm solutions. Thus, we explore an alternative path to explain the why DG algorithm succeeds in certain situations.
>
>    Most of the existing works that analyze noise robustness or the generalization properties of the DG algorithms are done in expectation, which effectively have **infinite** samples. In contrast, we choose to explain the phenomenon observed with **finite** samples. We think it is more valuable to perform finite-sample analysis because in practice the model can overfit and memorize noisy samples. Consequently, certain desirable properties may not always hold if only analyzed in expectation. However, finite-sample analysis for DG algorithms is challenging, and we do not have a formal theorem yet. Thus, we have taken a more intuitive approach by analyzing the gradients of IRM and V-REx algorithms. To the best of our knowledge, despite its simplicity, the gradient analysis and its insights for these DG algorithms has been overlooked in the past. We hope this would shed light on future studies of the internal working of DG algorithms under a more practical finite-sample setting.

---

> ### Author Response · Authors · 2023-11-18
> **Author Response to Reviewer qd9a (Part 2)**
>
> 4. > ... In page 4, how comes "the classifier become either $w_{spu}$ or $w_{inv}$"? ...
>
>    We would like to clarify that $w^{(inv)}$ stands for the classifier that uses the invariant and nuisance features. Similarly, $w^{(spu)}$ is for spurious and nuisance features. It is not necessary for the classifier to choose between $w^{(spu)}$ or $w^{(inv)}$. In practice, it is more likely to be a mix of both. The main goal of theorem 4.2 is to show that under that gap condition, there exists another classifier $w^{(spu)}$ that has less norm than $w^{(inv)}$ with high probability. This suffices to show the model will **not** converge to the invariant classifier and thus have more reliance on spurious features.
>
> 5. > The description of the Lemma C.1 is unclear ... it should be a conclusion that holds with high probability, but the lemma presents it deterministically.
>
>    Thank you for pointing this out. Yes, our main result is based on the probabilistic bound. We apologize for the incorrect wording and have added the term "with high probability" for both Lemma C.1 and Theorem 4.2.
>
> 6. >... Some of the symbols are not described (just mentioned) in this paper ...
>
>    We have added the mathematical definition of the symbols representing the classifier $w^{(spu)}$ and $w^{(inv)}$ in Appendix C.1. We will continue to improve the description of the symbols.
>
> 7. > ... In the experiment section, I recommend that the authors discuss the values of the weights' norms ...
>
>    Thank you for the suggestion. In Appendix D.3, we update the experiments on synthetic data and plot the norms of $w^{(spu)}$, $w^{(inv)}$, $w^{(spu)}\_{spu}$, $w^{(inv)}\_{inv}$ with respect to increasing noise levels. The results show that as the noise level goes up, $w^{(spu)}$ indeed tends to have a smaller norm than $w^{(inv)}$. At the same time, the change in norm gap between $w^{(spu)}\_{spu}$ and $w^{(inv)}\_{inv}$ is relatively small. Moreover, the classifier $w$ that uses all features also gets closer to $w^{(spu)}$. The empirical evidence is consistent with the implications of Theorem 4.2. However, for real-world image datasets, we are unable to show the norm comparison since we cannot be sure about which features are invariant or spurious.
>
>
> **Questions**:
> 1. > In all tables in experiments, the authors should add references to Mixup, GroupDRO, IRM, VREx.
>
>    As adding references for these tables in ICLR format will break the current typeset, we have opted for writing the baselines in the main paper and reference them there.
>
> 2. > The author should provide another theorem to illustrate that the classifier with a smaller norm becomes more favored by the model.
>
>    In Appendix C.2, we have added two formal theorems shown by [5,6] related to the implicit regularization effect of (stochastic) gradient descent. In particular, [5] proved that logistic regression converges in the direction of the max-margin classifier, which is equivalent to the minimum-norm classifier given the right constraints. Thus, we can conclude that the model has the inductive bias of learning models with smaller norms.
>
> 3. > The authors should provide further explanations on the regularization term in the practical surrogate.
>
>    The goal of IRM [1] is to obtain classifiers that are simultaneously optimal across all training environments, but this induces a challenging bilevel optimization problem. As zero gradient is necessary for reaching the local minimum, IRM has the practical surrogate that minimizes the norm of the gradient to encourage reaching the local minimum for all training environments. Since our goal is to understand how the DG algorithms work in practice, we only analyze the practical surrogate of IRM, which is the de facto implementation across standard benchmarks. This has been updated at the beginning of Appendix B.1.
>
> We hope that our justifications and discussions have addressed your concerns and will improve your opinion of our work.
>
> [1] Arjovsky, M., Bottou, L., Gulrajani, I., & Lopez-Paz, D. (2019). Invariant risk minimization. arXiv preprint arXiv:1907.02893.
>
> [2] Lyu, K., & Li, J. (2019). Gradient descent maximizes the margin of homogeneous neural networks. arXiv preprint arXiv:1906.05890.
>
> [3] Nagarajan, V., Andreassen, A., & Neyshabur, B. (2020). Understanding the failure modes of out-of-distribution generalization. arXiv preprint arXiv:2010.15775.
>
> [4] Sagawa, S., Raghunathan, A., Koh, P. W., & Liang, P. (2020). An investigation of why overparameterization exacerbates spurious correlations. In International Conference on Machine Learning (pp. 8346-8356). PMLR.
>
> [5] Soudry, D., Hoffer, E., Nacson, M. S., Gunasekar, S., & Srebro, N. (2018). The implicit bias of gradient descent on separable data. The Journal of Machine Learning Research, 19(1), 2822-2878.
>
> [6] Zhang, C., Bengio, S., Hardt, M., Recht, B., & Vinyals, O. (2021). Understanding deep learning (still) requires rethinking generalization. Communications of the ACM, 64(3), 107-115.

---

> ### Author Response · Authors · 2023-11-22
> **Gentle Reminder**
>
> Dear reviewer qd9a, we would like to extend our thanks for your thoughtful review. If you have any remaining questions or concerns that were not covered in our previous response, we are glad to discuss them further.

---

### Official Review · Reviewer_4qg5 · 2023-10-30

**Soundness:** 2 fair
**Presentation:** 3 good
**Contribution:** 3 good
**Rating:** 6
**Confidence:** 4

**Summary:**

This study offers a thorough exploration, both theoretically and empirically, to ascertain the situations where Domain Generalization (DG) algorithms outperform Empirical Risk Minimization (ERM) counterparts. The findings reveal that DG algorithms exhibit greater resilience in the presence of label noise during subpopulation shifts. On the other hand, ERM approaches tend to be susceptible to capitalizing on spurious correlations, particularly in overparameterized models. The study backs these observations with a blend of theoretical insights and empirical evidence.

**Strengths:**

S1 -The paper upholds a commendable level of clarity in its exposition, effectively conveying intricate ideas in an easily comprehensible fashion. Its tone is suitably professional, aligning with the subject matter, which contributes to a satisfying reading experience. The structural organization into subsections facilitates navigation and swift access to specific information. Furthermore, the notation employed is consistently precise, enabling readers to grasp the mathematical elements of the paper with ease. Notably, the theoretical statements are  thoughtfully elucidated through illustrative examples and substantiated by empirical findings.

S2 - The claims presented in the paper are substantiated through a combination of theoretical and empirical evidence.

S3 - The paper's commitment to reproducibility is highly commendable. The detailed and transparent presentation of the experimental setup, data sources, and code availability significantly enhances the reliability and trustworthiness of the research findings.

S4 - This paper thoroughly explores different scenarios for empirical validation and uses a diverse set of datasets (i.e, classification tasks). The authors have selected datasets that are commonly featured in the existing literature on this research topic.

**Weaknesses:**

W1 - The section discussing related work appears somewhat limited. Domain generalization is a machine learning technique designed to train models for effective performance on unfamiliar data originating from multiple domains or distributions. As such, it covers a broad and diverse research landscape, accommodating various scenarios and types of data shift. To enhance reader comprehension and provide a more thorough context, it would be beneficial for the authors to initiate with a comprehensive overview of domain generalization before delving into the specific scenario they focus on, which involves challenges such as label noise, spurious correlation, and subpopulation shifts. This approach can mitigate potential misunderstandings and offer a more holistic understanding for the readers.

W2 -  The paper primarily presents an empirical analysis, and it is commendable that the authors have incorporated a diverse set of datasets, which enhances the generalizability of their findings. However, it's worth noting that the diversity observed in the choice of datasets is not reflected in the selection of Domain Generalization (DG) methods considered for the analysis. In order to strengthen the general statements proposed in the paper, it would be valuable to expand the range of DG algorithms under examination. This broader inclusion of methods can further validate and reinforce the claims made in the paper, offering a more comprehensive view of the research landscape.

W3 - The experiments of the paper are limited to tasks related to image classification, with no inclusion of other types of data, such as tabular data. This focus on image classification tasks allows the authors to delve deeply into this specific domain and gain insights relevant to this context. However, it's important to note that the findings and conclusions may not be directly applicable to other data types or domains, and this limitation should be kept in mind when interpreting the results.

W4 - Section 5 only provides a description of IRM, but it would be valuable to have a more extensive discussion about other methods as well.

**Questions:**

Q1 - Can your theoretical framework comprehensively explain all domain generalization methods that address subpopulation shift and spurious correlations, or are there any inherent limitations?

Q2 - What motivated the selection of the methods used in the experimental section? Why were these particular methods chosen?

---

> ### Author Response · Authors · 2023-11-18
> **Author Response to Reviewer 4qg5 (Part 1)**
>
> We thank you for acknowledging our theoretical and empirical contributions to domain generalization and providing insightful suggestions. We address your comments as follows:
>
> **Summary:**
> - We have added empirical results for four more related algorithms and one NLP dataset to improve the generality of the results.
> - We provide additional analysis of the limitations of our approach and more discussion about why the DG algorithms are selected.
> - We have revised the paper accordingly with updates highlighted in blue.
>
> **Weaknesses:**
> 1. > W1: ... To enhance reader comprehension and provide a more thorough context, it would be beneficial for the authors to initiate with a comprehensive overview of domain generalization before delving into the specific scenario ...
>
>     We thank you and agree with your suggestion. As we have already provided a comprehensive overview of the domain generalization problem in Section 3, we will move Section 3 ahead of Section 2 to improve the flow. We have also improved the writing of related work by adding more details.
>
> 2. > W2: ...  In order to strengthen the general statements proposed in the paper, it would be valuable to expand the range of DG algorithms under examination ...
>
>     We agree that empirically examining more DG methods would improve our understanding and have added four more algorithms: CORAL [4] (domain adaption), IB_IRM [1] (DG), Fish [3] (DG), and Fishr [2] (DG). More details about the algorithms and results can be found in Appendix G.2.
>     Due to the time limit, we evaluate these models on CMNIST, Waterbirds+, and PACS datasets, from which we can see the same empirical trend presented in the main paper still holds. For CMNIST, certain DG algorithms like Fishr and IB_IRM have the advantage over ERM. However, for real-world datasets, none of these algorithms clearly outperform ERM even under label noise setting. We believe that for these algorithms, the same pattern should also generalize to other datasets.
>
>     As our experiments are extensive, to examine the algorithms further on all datasets, it would require training 1260 models for each noise level, which is quite expensive. Therefore, we only pick out a few representative datasets and influential algorithms for comparisons due to limited compute and space.
>
> 3. > W3: The experiments of the paper are limited to tasks related to image classification, with no inclusion of other types of data, such as tabular data. This focus on image classification tasks allows the authors to delve deeply into this specific domain and gain insights relevant to this context. However, it's important to note that the findings and conclusions may not be directly applicable to other data types or domains, and this limitation should be kept in mind when interpreting the results.
>
>     To incorporate more data types into our analysis, we perform empirical evaluation on CivilComments, a textual dataset for toxicity classification. The task description and results are added in Appendix G.1. In this NLP task, Mixup cannot be directly applied due to textual input. For other algorithms, GroupDRO has the lead in the noise-free setting, and V-REx exhibits slightly better noise-robustness. However, ERM still performs competitively compared to other algorithms even under label noise setting, which is consistent with the existing empirical observations.
>
> 4. > W4: Section 5 only provides a description of IRM, but it would be valuable to have a more extensive discussion about other methods as well.
>
>     Other than IRM, we have also included the analysis for the V-REx algorithm in Appendix B.2 and show that it has a similar property of oscillating between gradient descent and ascent for some data points, though with a slightly different mechanism. Furthermore, we have discussed in Section 6 about ERM, Mixup, and GroupDRO essentially sharing the same ERM objective and thus not having the noise robustness property. We choose IRM and V-REx as the two representative invariance learning algorithms because of their effectiveness and existing theoretical guarantee in expectation. We will discuss the challenges in extending our results to other DG approaches below in response to Q1.

---

> ### Author Response · Authors · 2023-11-18
> **Author Response to Reviewer 4qg5 (Part 2)**
>
> **Questions**
> 1. > Q1 - Can your theoretical framework comprehensively explain all domain generalization methods that address subpopulation shift and spurious correlations, or are there any inherent limitations?
>
>     Our first theoretical analysis in Section 4 on the failure condition of ERM due to label noise and spurious correlations is applicable to both subpopulation shift and domain shift in linear settings. However, in practice, when the feature extractors are non-linear and both the invariant and spurious features may not be perfectly extracted by the model, our result from linear settings may not be directly applicable. Therefore, we perform extensive empirical study to verify our hypothesis in synthetic and real-world non-linear settings in Sections 6.2 and 6.3, respectively. We can observe that the performance of ERM degrades severely when label noise is introduced for synthetic and real-world subpopulation shifts. However, the issue is less severe for domain shifts.
>
>     Our second analysis in Section 5 addresses the noise robustness of two invariance learning algorithms: IRM and V-REx, but not all domain generalization methods can be theoretically proven to possess such a desirable property. Various existing DG algorithms have different working mechanisms and it is challenging to generalize to all. For example, Fish [4] and Fishr [2] align the sample gradient from different environments, so that more assumptions are needed to examine the effect of gradient alignments on noise robustness or other properties. Moreover, similarities among DG algorithms may be drawn in expectation with infinite samples, but we focus on a more practical finite-sample perspective which adds more difficulty. We have added the relevant discussion in the revised paper.
>
> 2. > Q2 - What motivated the selection of the methods used in the experimental section? Why were these particular methods chosen?
>
>     For ERM-based algorithms, Mixup is a simple yet effective approach that has been widely studied from both empirical and theoretical perspectives.
>
>     For DG algorithms, we choose a few most influential and well-cited algorithms with theoretical guarantees for comparisons. These algorithms also constitute the baselines for many other DG algorithms. In particular:
>     - IRM is the first algorithm proposed to learn invariant/causal predictors from multiple environments with deep learning.
>     - V-REx extends IRM with a simplified objective function that addresses covariate shift in the input distribution. It is also theoretically analyzed for homoskedastic environments (consistent noise level).
>     - GroupDRO is one of the first algorithms proposed to specifically address subpopulation shifts by minimizing the worst-group risk directly. By utilizing the group information effectively, it has been considered the gold standard amongst many subsequent works that try to deal with subpopulation shifts without group labels.
>
>     We have added detailed descriptions of these algorithms in Appendix E.2.
>
> We hope that the additional insights provided in our responses and the revised paper can further improve your opinion of our paper.
>
> **References:**
>
> [1] Ahuja, K., Caballero, E., Zhang, D., Gagnon-Audet, J. C., Bengio, Y., Mitliagkas, I., & Rish, I. (2021). Invariance principle meets information bottleneck for out-of-distribution generalization. Advances in Neural Information Processing Systems, 34, 3438-3450.
>
> [2] Rame, A., Dancette, C., & Cord, M. (2022, June). Fishr: Invariant gradient variances for out-of-distribution generalization. In International Conference on Machine Learning (pp. 18347-18377). PMLR.
>
> [3] Sun, B., & Saenko, K. (2016). Deep coral: Correlation alignment for deep domain adaptation. In Computer Vision–ECCV 2016 Workshops: Amsterdam, The Netherlands, October 8-10 and 15-16, 2016, Proceedings, Part III 14 (pp. 443-450). Springer International Publishing.
>
> [4] Shi, Y., Seely, J., Torr, P. H., Siddharth, N., Hannun, A., Usunier, N., & Synnaeve, G. (2021). Gradient matching for domain generalization. arXiv preprint arXiv:2104.09937.

---

> > ### Comment · Reviewer_4qg5 · 2023-11-20
> > **Response to authors**
> >
> > Thank you very much for your detailed response to my concerns.
> >
> > Based on your response to my first question and as highlighted by reviewers qd9a and TZVP, the assumption of linear separability holds significant importance in your analysis and the subsequent findings and conclusions. While it serves as a valuable tool for theoretical examination, it is imperative that the paper explicitly acknowledges its confinement to a specific scenario and clearly articulates the associated limitations.
> >
> > Therefore, for the moment I will keep my score.

---

> > > ### Author Response · Authors · 2023-11-22
> > > **Author Response to Reviewer 4qg5 (Part 3)**
> > >
> > > Thank you for the swift response and critical suggestions. We acknowledge these limitations and hope that the future research can progressively improve upon our current work. Therefore, we have added a brief overview about the limitations in Section 8 of the main paper and a more comprehensive discussion regarding three limitations in Appendix H. Hope this helps!
> > >
> > > Moreover, in case you have missed our new experimental results in Appendix for the previous response. We put them here for a better reference.
> > >
> > > Table 1: Worst-group accuracy (%) for CivilComments (NLP domain).
> > >
> > > | **Algorithm/$\eta$** | **0**               | **0.1**             | **0.25**            | **Avg**  |
> > > |-|-|-|-|-|
> > > | ERM                       | $89.8\pm_{0.4}$          | $67.9\pm_{0.4}$          | $55.4\pm_{0.1}$          | $71.0$          |
> > > | IRM                       | $89.2\pm_{0.2}$          | $66.9\pm_{0.4}$          | $54.2\pm_{0.3}$          | $70.1$          |
> > > | GroupDRO                  | $\textbf{91.9}\pm_{0.2}$ | $\textbf{69.7}\pm_{0.3}$ | $55.4\pm_{0.1}$          | $\textbf{72.3}$ |
> > > | VREx                      | $89.6\pm_{0.3}$          | $69.0\pm_{0.4}$          | $\textbf{56.0}\pm_{0.3}$ | $71.6$          |
> > >
> > >
> > > Table 2: Test accuracy (%) on CMNIST.
> > >
> > > | **Algorithm/$\eta$** | **0**               | **0.1**             | **0.25**            | **Avg**  |
> > > |-|-|-|-|-|
> > > | ERM                       | $97.1\pm_{0.1}$          | $77.8\pm_{1.9}$          | $39.5\pm_{1.2}$          | $71.5$          |
> > > | Mixup                     | $\textbf{97.3}\pm_{0.2}$ | $82.3\pm_{0.9}$          | $24.7\pm_{1.9}$          | $68.1$          |
> > > | GroupDRO                  | $\textbf{97.3}\pm_{0.1}$ | $84.5\pm_{1.7}$          | $48.3\pm_{1.0}$          | $76.7$          |
> > > | IRM                       | $96.8\pm_{0.3}$          | $\textbf{86.5}\pm_{3.8}$ | $\textbf{73.2}\pm_{9.1}$ | $\textbf{85.5}$ |
> > > | VREx                      | $96.7\pm_{0.1}$          | $84.1\pm_{1.8}$          | $57.8\pm_{2.0}$          | $79.5$          |
> > > | CORAL                     | $97.1\pm_{0.2}$          | $77.3\pm_{1.4}$          | $37.2\pm_{0.7}$          | $70.5$          |
> > > | IB\_IRM                   | $96.5\pm_{0.5}$          | $83.7\pm_{4.4}$          | $71.3\pm_{5.8}$          | $83.8$          |
> > > | Fish                      | $97.2\pm_{0.2}$          | $78.8\pm_{0.6}$          | $37.7\pm_{1.3}$          | $71.2$          |
> > > | Fishr                     | $\textbf{97.3}\pm_{0.0}$          | $82.0\pm_{1.3}$          | $67.6\pm_{9.4}$          | $82.3$          |
> > >
> > > Table 3: Worst-group accuracy (%) for Waterbirds+.
> > >
> > > | **Algorithm/$\eta$** | **0**               | **0.1**             | **0.25**            | **Avg**  |
> > > |-|-|-|-|-|
> > > | ERM                       | $81.5\pm_{0.4}$          | $64.6\pm_{0.5}$          | $50.4\pm_{0.9}$          | $65.5$          |
> > > | IRM                       | $78.5\pm_{0.8}$          | $63.3\pm_{0.5}$          | $47.8\pm_{1.8}$          | $63.2$          |
> > > | GroupDRO                  | $\textbf{82.6}\pm_{0.4}$ | $65.6\pm_{0.8}$          | $51.2\pm_{0.7}$          | $\textbf{66.5}$ |
> > > | Mixup                     | $79.8\pm_{0.9}$          | $61.8\pm_{2.0}$          | $51.3\pm_{1.5}$          | $64.3$          |
> > > | VREx                      | $78.3\pm_{0.3}$          | $\textbf{66.6}\pm_{0.9}$ | $51.9\pm_{0.5}$          | $65.6$          |
> > > | CORAL                     | $80.8\pm_{0.6}$          | $64.3\pm_{1.8}$          | $\textbf{53.5}\pm_{1.8}$ | $66.2$          |
> > > | IB\_IRM                   | $74.5\pm_{1.2}$          | $60.8\pm_{1.5}$          | $48.6\pm_{1.4}$          | $61.3$          |
> > > | Fish                      | $82.3\pm_{0.3}$          | $66.4\pm_{0.4}$          | $49.8\pm_{2.4}$          | $66.2$          |
> > > | Fishr                     | $79.4\pm_{0.7}$          | $62.1\pm_{0.5}$          | $\textbf{53.5}\pm_{1.3}$ | $65.0$          |
> > >
> > > Table 4: Cross-test accuracy (%) for PACS.
> > >
> > > | **Algorithm/$\eta$** | **0.1**         | **0.25**        | **Avg** |
> > > |-|-|-|-|
> > > | ERM                           | $82.0\pm_{0.5}$          | $74.5\pm_{0.6}$          | $78.3$             |
> > > | IRM                           | $80.4\pm_{1.2}$          | $71.0\pm_{1.9}$          | $75.7$             |
> > > | GroupDRO                      | $82.4\pm_{0.3}$          | $74.7\pm_{0.4}$          | $78.5$             |
> > > | Mixup                         | $\textbf{83.6}\pm_{0.1}$ | $75.2\pm_{0.9}$          | $\textbf{79.4}$    |
> > > | VREx                          | $81.4\pm_{0.2}$          | $73.5\pm_{0.7}$          | $77.4$             |
> > > | CORAL                         | $82.6\pm_{0.4}$          | $75.9\pm_{0.3}$          | $79.2$             |
> > > | IB\_IRM                       | $77.0\pm_{2.5}$          | $67.8\pm_{3.5}$          | $72.4$             |
> > > | Fish                          | $83.1\pm_{0.1}$          | $\textbf{77.0}\pm_{0.7}$ | $\textbf{80.1}$    |
> > > | Fishr                         | $81.7\pm_{0.4}$          | $74.5\pm_{0.5}$          | $78.1$             |

---

### Official Review · Reviewer_jUs6 · 2023-10-30

**Soundness:** 2 fair
**Presentation:** 3 good
**Contribution:** 2 fair
**Rating:** 3
**Confidence:** 3

**Summary:**

This paper explores whether the Domain Generalization (DG) algorithm outperforms the classic Empirical Risk Minimization (ERM) algorithm in the presence of labeled noise, and why.

**Strengths:**

S1. The writing expression of this paper is relatively clear, but it is still not standardized enough, such as Eqn 1 should be written as Eqn (1).

S2. The research motivation of this paper is to explore the effectiveness of DG compared to ERM under labeled noise settings. This is positive for the study of DG, after all, there is no clear empirical evidence that the existing DG algorithms perform the classic ERM across standard benchmarks.

**Weaknesses:**

W1. The failure of validation on real data is pessimistic, which seriously reduces the importance of the settings discussed in this paper, as real data does not fit well with simple noise settings.

W2. The main theoretical results of this paper have poor readability. The conclusion described in Theorem 4.2 is not very intuitive. It is difficult to associate this principle with the main contributions described in the abstract of this paper.

**Questions:**

Q1. I would like to know the relationship between Theorem 4.2 and "Specifically, our finite-sample analysis reveals that label noise exacerbates the effect of spurious correlations for ERM, undermining generalization. Conversely, we illustrate that DG algorithms exhibit implicit label-noise robustness during finite-sample training even when spurious correlation is present." in the abstract? How does this theorem reflect label noise?

---

> ### Author Response · Authors · 2023-11-18
> **Author Response to Reviewer jUs6 (Part 1)**
>
> We thank your comments and acknowledging our positive contribution to domain generalization. We would like to address your comments as follows:
>
> **Summary**:
> - We discuss the importance of our work.
> - We provide clearer explanations of the relationship between Theorem 4.2 and our main contributions. We revise the related paragraphs in Section 4.1.
>
> **Weaknesses:**
>
> 1. > W1. The failure of validation on real data is pessimistic, which seriously reduces the importance of the settings discussed in this paper, as real data does not fit well with simple noise settings.
>
>     As our goal is to objectively study domain generalization, the result on real-world data with label noise (ERM-based methods performing competitively with DG algorithms) is not entirely pessimistic. We have demonstrated the consistency between theory and practice and the benefit of DG algorithms with experiments in coloredMNIST dataset. Moving on to real-world cases, the data is more complex and the features are more difficult to extract. We acknowledge that there is a gap between our theory and practice. However, our extensive empirical results on real-world data help researchers to develop an understanding and critically review the existing DG approaches. Moreover, we have elaborated in Section 7 about the potential reasons why the same patterns from synthetic datasets are not observed in real-world cases. We believe that our theoretical and empirical analyses provide insights from a new perspective and could drive new research advances in this area.
>
>     Although real-world data are affected by more complicated noise, label noise is an important issue that is prevalent across domains. We believe that it is common to work with some form of simulated noise in the beginning, such as the uniform label noise used in this work.
>
> 2. > W2. The main theoretical results of this paper have poor readability ... It is difficult to associate this principle with the main contributions described in the abstract of this paper.
>
>     We have improved the writing in Section 4.1 of the revised paper with blue-colored text and hope that could help to clarify the connection between Theorem 4.2 and our contributions. We also provide more detailed explanations in response to your Q1 below.
>
> **Questions:**
> - > Q1.  I would like to know the relationship between Theorem 4.2 and ... How does this theorem reflect label noise?
>
>   For "*Specifically, our finite-sample analysis reveals that label noise exacerbates the effect of spurious correlations for ERM, undermining generalization*", the overall rationale is that: label noise causes the spurious classifier to have less norm than the invariant classifier so that the model will rely more on spurious correlation. The rationale can be obtained with two steps:
>   - Theorem 4.2 shows that having a more severe spurious correlation $\gamma$ and higher noise level $\eta$ both make the condition in Theorem 4.2 easier to satisfy. To interpret this, as the **noise level** of the label increases, the term $n(1 − \gamma)(1 − 2\eta)C$ on the right-hand side decreases, but the norm gap on the left-hand side of the inequality remains unchanged. Thus, the "if" condition is easier to satisfy, causing the spurious classifier to have less norm than the invariant one.
>   - We have mentioned the minimum-norm bias of gradient descent (GD) in Section 4.1. This is also known as the implicit regularization effect [1,2]. As both types of classifiers are capable of reaching 0 training loss, the spurious classifier with less norm (less complex) is preferred by DG. Thus, the actual model we obtain is not purely invariant but utilizes spurious features, which hurts generalization.
>
>   We restrict the consequence of having label noise to "exacerbating the effect of spurious correlations" because Theorem 4.2 also relies on the degree of spurious correlations. When the spurious correlation is high, having label noise will make it worse. In particular, for our synthetic linear and non-linear experiments, high spurious correlation may be less of an issue in a noise-free setting, but when combined with label noise, the generalization performance degrades severely.
>
>   For "*Conversely, we illustrate that DG algorithms exhibit implicit label-noise robustness during finite-sample training even when spurious correlation is present*", in Section 5, we analyze the gradient of IRM (V-REx is also available in Appendix B.2), and show that it is more robust to memorizing label noise. In Section 6.2, on CMNIST dataset, we empirically show that IRM and V-REx algorithms perform well even after injecting label noise. However, ERM and Mixup perform worse than random guessing, indicating that spurious correlations are being exploited, despite searching over 60 sets of hyperparameters.
>
>
> We sincerely hope our responses above have clarified your doubts about the connection between our contributions and Theorem 4.2 and will improve your opinion of our paper.

---

> ### Author Response · Authors · 2023-11-18
> **Author Response to Reviewer jUs6 (Part 2, References)**
>
> **References:**
>
> [1] Soudry, D., Hoffer, E., Nacson, M. S., Gunasekar, S., & Srebro, N. (2018). The implicit bias of gradient descent on separable data. The Journal of Machine Learning Research, 19(1), 2822-2878.
>
> [2] Zhang, C., Bengio, S., Hardt, M., Recht, B., & Vinyals, O. (2021). Understanding deep learning (still) requires rethinking generalization. Communications of the ACM, 64(3), 107-115.

---

> ### Author Response · Authors · 2023-11-22
> **Gentle Reminder**
>
> Dear reviewer jUs6, we appreciate you spending valuable time reviewing our work. We wish to confirm if our clarifications above have resolved your main concerns and we are glad to clarify any remaining doubts.

---

### Official Review · Reviewer_TZVP · 2023-10-30

**Soundness:** 2 fair
**Presentation:** 3 good
**Contribution:** 2 fair
**Rating:** 5
**Confidence:** 3

**Summary:**

As many previous studies have numerically demonstrated, no domain generalization method clearly outperforms the empirical risk minimization in general. This study investigates when and why DG methods better generalize than the empirical risk minimization and vice versa, through the lens of label-noise and subpopulation shifts. Particularly, the authors demonstrate the empirical risk minimization's tendency to learn spurious correlations (or domain-specific features) rather than invariant features for overparameterized models determined by both degrees of spurious correlation and label noise. Moreover, the authors also investigated that some domain generalization methods can learn invariant features over spurious correlations, resulting in better generalizability in the presence of noisy labels. Extensive numerical experiments were provided.

**Strengths:**

- Theoretical analysis on when and why the domain generalization methods perform better or worse than the empirical risk minimization has rarely been studied. This paper provides a concrete and interesting one.
- Well-written and easy to follow. Assumptions for the analysis have been made clear.

**Weaknesses:**

- The analyses provided in this study are based on a linear setting, assuming that we have disjoint sets of invariant, spurious, and nuisance predictors. This seems to be a reasonable assumption for theoretical analysis. However, in real-world cases, we might not be able to have such disjoint sets of predictors. For example, in the computer vision tasks given in the experimental study, it is not straightforward that we have such predictors unless a neural network learns such appropriate representations, which I think is hardly possible.
- Even for tabular data, we might need a proper transformation to have such ideal sets of predictors.
- So, such difficulty in having an appropriate representation might be responsible for the failure of the noise robustness to translate to better generalizability in the experimental study.
- In short, the theory is sound, however, the conclusion from the experimental study is not fully convincing, and seems to need further exploration.

**Questions:**

- I think a simpler simulation scenario, such as the linear case given in Section 4.1, might be more appropriate to demonstrate the theory. As mentioned in the weaknesses, the computer vision scenarios presented in the experimental study, require an ideal feature extractor that can provide invariant, spurious, and nuisance features. However, there is no guarantee that such representations were learned.
- The authors might need an ideal feature extractor that provides the disjoint sets of invariant, spurious, and nuisance predictors or a proxy of such extractor to demonstrate the hypothesis.
- It seems like the overall problem setting is also relevant to the fairness problem. Is there any relevant study from algorithmic fairness literature that the authors know of?
- Is it possible to extend the discussion in Section 4, which is focused on subpopulation shift, to analyze the domain shift? Particularly, for the cases where the ERM or DG might fail.

---

> ### Author Response · Authors · 2023-11-18
> **Author Response to Reviewer TZVP (Part 1)**
>
> We thank you for the insightful comments and recognizing our contribution to studying the advantages of domain generalization methods.
>
> **Summary:** we have addressed:
> - The practicality, generality, and limitations of us assuming the availability of disjoint features.
> - How representation learning can be responsible for the experimental observations in real-world datasets.
> - Related work to fairness.
> - Potential extension to domain shifts.
>
> Detailed responses to your comments are as follows:
>
> **Weaknesses:**
> - > **W1, W2, Q1, Q2**: ... assuming that we have disjoint sets of invariant, spurious, and nuisance predictors. This seems to be a reasonable assumption for theoretical analysis. However, in real-world cases, we might not be able to have such disjoint sets of predictors ... Even for tabular data, we might need a proper transformation to have such ideal sets of predictors.
>
>   We believe your major concern for Weakness 1,2 and Question 1,2 is the practicality of our assumption on having disjoint sets of invariant, spurious, and nuisance features, which require ideal feature extractors. We would like to argue that our analyses in Section 4 have broader implications:
>   1. For *linear* cases, though not explicitly stated, our theoretical analysis in Theorem 4.2 generalizes to rotational feature transformation. To see this: logistic regression and SVMs have the rotational invariance property [6], which means that training on data with *original* features or *rotated* features effectively yield the same decision boundaries.
>   This natural extension implies that the invariant, spurious, and nuisance features do not have to be in distinct dimensions, as long as they are orthogonal to each other. To further strengthen the idea, we extend our synthetic experiments by transforming the input features with random projection matrices onto the same dimension. We observe almost exactly the same trends in comparison to the case without random projection. We have included the discussions and experimental results in Appendix D.2, which is consistent with Figure 1 in the main paper. We believe this also addresses the concern for tabular data if the transformation is linear.
>   2. Similar setting with linear models learning from orthogonal invariant, spurious, and nuisance features has been also made in prior theoretical works [5,8].
>   3. For real-world tasks with *non-linear* feature transformations, Due to the complexity of deep learning models, we agree that it is hard to identify which and whether features have been extracted.
>   However, we would like to highlight that the main objective of Section 4 is to show how label noise prevents ERM from utilizing invariant features. What we have demonstrated is the failure mode when invariant and spurious features can be extracted. Now, let's briefly consider two other scenarios where the ideal feature extractor does not exist:
>       - Spurious features are not fully extracted: This can be viewed as having a weaker spurious correlation and our results still apply.
>       - Invariant features are not fully extracted: The model has to exploit more spurious correlations and nuisance features, so it already fails to generalize.
>
>       We believe that showing the failure mode of ERM under such an ideal situation encompasses some other failure modes without the ideal feature extractors. From this perspective, the assumption may not be overly restrictive. However, we agree such an assumption may affect the analysis of the benefit of noise robustness of DG algorithms, which we discuss below for W3.
>
> - > **W3**: So, such difficulty in having an appropriate representation might be responsible for the failure of the noise robustness to translate to better generalizability in the experimental study.
>
>   We thank you for pointing out the insight and agree that representation learning is an important aspect of domain generalization. If such representations cannot be extracted in practice and only spurious features are picked up by the model, then no algorithms would work perfectly here. The phenomenon of DG algorithms succeeding for synthetic data but being on par with ERM for real-world data could potentially be caused by the representations learned. In particular, it might be the case that CMNIST has easier invariant features, but the real-world dataset does not, causing all models to fail to generalize. Currently, the standard evaluation of DG algorithms is still based on ResNet-50 pretrained on ImageNet. With more powerful feature extractors, it may be possible to see the expected trend as in the synthetic case, which is left for future work. We acknowledge that assuming those features are accessible by the model in the theoretical setup is one of the limitations that cause the gap from theory to practice, but we believe our analyses provide valuable insights from a new perspective to the DG community.

---

> ### Author Response · Authors · 2023-11-18
> **Author Response to Reviewer TZVP (Part 2)**
>
> - > **W4**: ... the conclusion from the experimental study is not fully convincing, and seems to need further exploration.
>
>   The key message from our extensive empirical study is that DG algorithms work well for synthetic datasets with label noise due to their implicit noise robustness, but this is still not good enough to convincingly outperform ERM in real-world scenarios. We also provide our hypotheses and discussions in Section 7. As the aim of the paper is to demonstrate understanding and provide insights for DG problems, we hope our results can shed light on future works that take the implicit noise robustness into account.
>
> **Questions:**
> - > **Q1, Q2**: .... The authors might need an ideal feature extractor that provides the disjoint sets of invariant, spurious, and nuisance predictors ...
>
>   Please refer to our response to weaknesses 1,2 and 3.
>
> - > **Q3**: It seems like the overall problem setting is also relevant to the fairness problem. Is there any relevant study from algorithmic fairness literature that the authors know of?
>
>   Algorithmic fairness aims to obtain predictors that satisfy certain criteria of fairness, such as demographic parity [3] or equalizing odds [4]. As discussed in [2], some fairness criteria can be related to the objectives used by DG. While DG focuses more on learning invariant representation that generalizes across environments, fairness emphasizes removing the bias, though fundamentally there are many similarities. Based on our literature review so far, from the perspective of label noise, the most similar work that we found is "*Fair classification with group-dependent label noise"*[9]. This work and its follow-ups show that when naively using fair constraints, having label noise for certain groups can degrade the accuracy for groups without label noise, such that the result becomes worse than ERM. However, it differs significantly from our work in two ways:
>   - Their noise setting is group-dependent, while ours is uniform without discrimination.
>   - They show that label noise is more harmful to fairness algorithms than ERM in expectation, while our finite-sample analysis and experiments show the opposite of that: label noise is harmful to ERM, but some DG algorithms can be more robust to that.
>
>   We believe that such discrepancy is caused by their group-dependent label noise setting and our uniform noise setting is comparably more general to all groups. We have also added the discussion in Section 2 for related works.
>
> - > **Q4**: Is it possible to extend the discussion in Section 4, which is focused on subpopulation shift, to analyze the domain shift? Particularly, for the cases where the ERM or DG might fail.
>
>   Overall, if we treat the domain-specific features as spurious features $x_{spu}$, our norm-based theoretical results about ERM relying on spurious features are actually applicable to domain shift. However, the concepts of majority group (w/ spurious correlation) and minority group (w/o spurious correlation), which we want to highlight for better clarity, are mostly discussed for subpopulation shift but uncommon for domain shift. Moreover, extending to domain shifts requires assuming the domain-specific features to have strong correlations with the labels, which may complicate the discussion.
>
>   Regarding the failure mode of DG algorithms in domain shifts, we currently do not have a good theoretical answer. Prior works have shown invariance learning algorithms require a sufficiently large number of environments [1,7], so having too few environments constitutes a failure mode for DG. Other than this, we have also briefly discussed in Section 7 the possibility of learning spurious features also being a non-trivial task.
>
> We hope that our clarifications have addressed your concerns and will improve your evaluation of our work by bringing out more insights.

---

> ### Author Response · Authors · 2023-11-18
> **Author Response to Reviewer TZVP (Part 3, References)**
>
> **References:**
>
> [1] Arjovsky, M., Bottou, L., Gulrajani, I., & Lopez-Paz, D. (2019). Invariant risk minimization. arXiv preprint arXiv:1907.02893.
>
> [2] Creager, E., Jacobsen, J. H., & Zemel, R. (2021, July). Environment inference for invariant learning. In International Conference on Machine Learning (pp. 2189-2200). PMLR.
>
> [3] Dwork, C., Hardt, M., Pitassi, T., Reingold, O., & Zemel, R. (2012, January). Fairness through awareness. In Proceedings of the 3rd innovations in theoretical computer science conference (pp. 214-226).
>
> [4] Hardt, M., Price, E., & Srebro, N. (2016). Equality of opportunity in supervised learning. Advances in neural information processing systems, 29.
>
> [5] Nagarajan, V., Andreassen, A., & Neyshabur, B. (2020). Understanding the failure modes of out-of-distribution generalization. arXiv preprint arXiv:2010.15775.
>
> [6] Ng, A. Y. (2004, July). Feature selection, L 1 vs. L 2 regularization, and rotational invariance. In Proceedings of the twenty-first international conference on Machine learning (p. 78).
>
> [7] Rosenfeld, E., Ravikumar, P., & Risteski, A. (2020). The risks of invariant risk minimization. arXiv preprint arXiv:2010.05761.
>
> [8] Sagawa, S., Raghunathan, A., Koh, P. W., & Liang, P. (2020). An investigation of why overparameterization exacerbates spurious correlations. In International Conference on Machine Learning (pp. 8346-8356). PMLR.
>
> [9] Wang, J., Liu, Y., & Levy, C. (2021, March). Fair classification with group-dependent label noise. In Proceedings of the 2021 ACM conference on fairness, accountability, and transparency (pp. 526-536).

---

> ### Author Response · Authors · 2023-11-22
> **Gentle Reminder**
>
> Dear reviewer TZVP, we thank you once again for your thorough review. We would like to ask if you have any other questions beyond our rebuttal above and we hope to clarify them in time during the discussion period.

---

### Author Response · Authors · 2023-11-23
**Rebuttal Summary**

For a better reading experience, we summarize the discussions during the rebuttal as follows:
Overall, our work aims to provide more theoretical understanding and critical empirical analyses to facilitate research in the area of generalization and distribution shifts.

- **Reviewer TZVP** appreciates our novelty in the effort of studying when and why DG algorithms outperform ERM, but is concerned about our assumption of having disjoint sets of invariant, spurious, and nuisance predictors. In response to this:
  - Apart from the assumption being common in prior theoretical works, we have argued that our analysis under such an assumption is sufficient to demonstrate yet another failure mode of ERM under the effect of both label noise and spurious correlations.
  - We recognize the limitation of not having such ideal sets of features in practice, which can be one of the reasons for observing ERM and DG performing similarly in real-world data.
- **Reviewer jUs6** acknowledges the positive impact of our research for DG but is concerned about the readability of our theoretical result and its connection to the claimed contributions in the abstract. We have improved the writing and explained with better clarity.
- **Reviewer 4qg5** commends the diversity and completeness of our empirical study, and suggests that it can be improved further by incorporating:
  - *Datasets from other modalities*: We extend our experiments to CivilComments in the textual domain.
  - *More DG algorithms*: We include 4 more algorithms for comparisons in a few representative datasets and observe similar trends. Thus, we believe our empirical results generalize to other algorithms in the same condition.
- **Reviewer qd9a** recognizes our main theoretical and empirical messages but is concerned about:
  - *Our assumption on the linear separability of invariant features*: We discuss the inclusiveness of assuming linear separability of invariant features only for noise-free data.
  - *The validity of the minimum-norm bias of gradient descent*: We justify the minimum-norm bias by providing existing theoretical references and more extensive discussions. Moreover, we empirically show that the norms of the classifiers behave as expected using synthetic data.
  - *A lack of formal theorem for noise robustness*: We discuss the usefulness of our gradient analysis and the challenges faced to address the finite-sample case.

We hope that this summary can facilitate the decision-making process. We thank the precious time of all reviewers and readers.

---

### Meta-Review · Area_Chair_Yo1k · 2023-12-05

**Metareview:**

The paper provides a new theoretical perspective on DG algorithm over ERM with respect to noise robustness. The main concern is about the linear separability assumption of invariant features. Also, there are concerns about the mismatch between the theoretical results and the empirical evidence. However, the overall response from the reviewers is positive and they appreciate the novelty of such analysis in this domain.

**Justification For Why Not Higher Score:**

The theoretical results of the paper are interesting, but have some limiting assumptions such as linear separability.

**Justification For Why Not Lower Score:**

The paper is borderline acceptable. The authors acknowledge the limitations of their work, but the study is still novel and interesting, even with a linear separability assumption.

---

### Decision · Program_Chairs · 2024-01-16

Accept (poster)